# LUMOS-1: ON AUTOREGRESSIVE VIDEO GENERATION WITH DISCRETE DIFFUSION FROM A UNIFIED MODEL PERSPECTIVE

**Hangjie Yuan**[1,2,3], **Weihua Chen**[1,2,†], **Jun Cen**[1,2,3], **Hu Yu**[1], **Jingyun Liang**[1,2]
**Shuning Chang**[1,2,3], **Zhihui Lin**[1,2,3], **Tao Feng**[4], **Pengwei Liu**[3], **Jiazheng Xing**[3]
**Hao Luo**[1,2], **Jiasheng Tang**[1,2], **Fan Wang**[1], **Yi Yang**[3]

[1]DAMO Academy, Alibaba Group, [2]Hupan Lab, [3]Zhejiang University, [4]Tsinghua University
[†] Corresponding author. `hj.yuan@zju.edu.cn`
Code and models: `https://github.com/alibaba-damo-academy/Lumos`

## ABSTRACT

Autoregressive large language models (LLMs) have unified a vast range of language tasks, inspiring preliminary efforts in autoregressive (AR) video generation. Existing AR video generators either diverge from standard LLM architectures, depend on bulky external text encoders, or incur prohibitive latency due to next-token decoding. In this paper, we introduce `Lumos-1`, an LLM-based unified model for AR video generation with efficient discrete diffusion. Firstly, to fit videos with LLMs, we identify that 1D RoPE is ill-suited for visual spatiotemporal correlation modeling, and while demonstrated to be useful, naive 3D RoPE exhibits imbalanced frequency spectra. Therefore, we propose MM-RoPE, which preserves the original textual RoPE while seamlessly accommodating video data with comprehensive frequency spectra and scaled 3D positions. Secondly, to fit the video data's nature and overcome the inefficiency of next-token decoding, we adopt a parallel and mask-based discrete diffusion with the intra-frame bidirectional and inter-frame causal attention masks. Based on this attention mask, we uncover the frame-wise loss imbalance issue caused by spatial information redundancy and propose Autoregressive Discrete Diffusion Forcing, which introduces temporal tube masking during training with a compatible inference-time masking policy to avoid quality degradation. Despite using only 48 GPUs for pre-training and fine-tuning, limited data and a discrete tokenizer, `Lumos-1` achieves results surpassing those of Show-o2 on GenEval, COSMOS-Video2World on VBench-I2V, and OpenSoraPlan on VBench-T2V.

## 1 INTRODUCTION

Autoregressive (AR) models have demonstrated significant advancement in the field of language processing OpenAI (2023); Touvron et al. (2023a); Ouyang et al. (2022); Bai et al. (2023); Liu et al. (2024a) by unifying diverse language tasks into one single framework and expanding the scale of large language models to unprecedented sizes. Following this trail of research, significant efforts have emerged in autoregressive visual generation Li et al. (2024a); Chang et al. (2023; 2022); Luo et al. (2024). By utilizing similar architectural and generation designs as those used in LLMs, there is substantial potential for advancing LLMs towards a unified model capable of both visual generation and understanding Wang et al. (2024c); Xie et al. (2024); Zhou et al. (2024); Tong et al. (2024).

Pioneering research efforts have been dedicated to instantiating AR visual generation, with a focus on key aspects, spanning improving autoregressive paradigm Pang et al. (2024); Tian et al. (2024); Chang et al. (2022); Luo et al. (2024), improving tokenizer format Li et al. (2024a) (*e.g.*, from discrete to continuous), and enhancing tokenizer capabilities Tang et al. (2024); Wang et al. (2024a); Shi et al. (2024). However, an AR video generation paradigm that is fully compatible with LLM-based unified models (*e.g.*, Chameleon Team (2024) and EMU3 Wang et al. (2024c)) remains underexplored. Preliminary attempts either exhibit architectural distinctions with LLM architectures (*e.g.*, NOVA Deng et al. (2024) and Phenaki Villegas et al. (2022)), rely on external text encoders (*e.g.*, LlamaGen Sun et al. (2024) and Fluid Fan et al. (2024)), or trail in generation efficiency due

to next-token prediction (*e.g.*, Loong Wang et al. (2024d)). *Therefore, the introduction of LLMs to visual generation motivates careful designs for modeling the spatiotemporal visual space.*

Adapting LLMs for video generation presents two fundamental challenges. *First*, the standard positional encoding in LLMs, such as 1D Rotary Position Embeddings (RoPE) Su et al. (2024), is inherently designed for sequential text and is ill-suited for modeling the complex 3D spatiotemporal correlations of video. *Second*, the standard autoregressive paradigm of next-token prediction, while effective for text, is notoriously inefficient for visual data and fails to properly model videos' unique properties: spatial bidirectionality within a frame and strong temporal causality across frames. Addressing these two challenges—RoPE representation and prediction paradigm—is critical for building an effective and efficient model for video generation.

To account for the spatiotemporal nature of videos using the RoPE technique, we start with systematic experiments and analysis (in Sec. 3.1) of extending 1D RoPE to a vanilla 3D RoPE Hong et al. (2022); Kong et al. (2024), revealing that 1D RoPE is far from optimal, and the incorporation of 3D RoPE facilitates autoregressive

Table 1: **Design comparison with other types of RoPE**.

| RoPE Type | Compatiable with Text RoPE | 3D Structure | Comprehensive Frequency Allocation | Strategic Scaling |
|---|---|---|---|---|
| M-RoPE | ✔ | ✔ | ✗ | ✗ |
| U-RoPE | ✔ | ✔ | ✗ | ✗ |
| IL-RoPE | ✗ | ✔ | ✗ | ✗ |
| VideoRoPE | ✔ | ✔ | ✗ | ✔ |
| HoPE | ✔ | ✔ | ✗ | ✔ |
| MM-RoPE | ✔ | ✔ | ✔ | ✔ |

generative learning. However, existing 3D RoPE still suffers from imbalanced frequency spectrum ranges for temporal and spatial modeling, as shown in Sec. 3.1. Building on this insight, we propose MM-RoPE, a new family of RoPE that better accommodates the structure of video data. As summarized in Tab. 1, the core enhancement lies in more distributed frequency allocations for more comprehensive context modeling, substantially improving the visual generation capability. Moreover, we design a principled scaling to the 3D positions, which improves vision-language modality balancing. Though MM-RoPE modifies RoPE for visual tokens, the RoPE for text tokens remains consistent with native LLMs, therefore preserving the language learning capabilities.

To account for the nature of videos (*i.e.*, spatial bidirectionality and temporal causality), we build our model upon the intra-frame bidirectional and inter-frame causal token dependency strategy, so that we can introduce the use of the efficient and parallel discrete diffusion Chang et al. (2022). Discrete diffusion requires training with masks Li et al. (2024a). However, the naive global random mask prediction leads to the loss imbalance issue, which means that masked visual tokens in the later frames tend to have much lower loss, impeding effective video modeling. We attribute this to videos' spatial information redundancy, as spatial information leakage allows the model to easily predict the masked token by attending to unmasked ones in previous frames. To address this, we propose Autoregressive Discrete Diffusion Forcing (AR-DF). The core of AR-DF training involves temporal tube masking Tong et al. (2022), which repeats a frame-level random mask pattern across the temporal axis. Such a design prevents masked tokens in later frames from learning a shortcut–copying previous ones. The core of AR-DF inference is the strategic inference-time masks that align with partial observation of history during training, enabling inference without degraded frame quality and motion.

Collectively, we propose `Lumos-1`, a model building on Llama Touvron et al. (2023b) and discrete diffusion to achieve AR video generation. Through a stage-wise training with GPU memory-friendly techniques, we pre-train `Lumos-1` *from scratch* on 60 million images and 10 million videos using only 48 GPUs, achieving performance comparable to EMU3 on GenEval, COSMOS-Video2World on VBench-I2V, and OpenSoraPlan on VBench-T2V.

Our contributions can be summarized as: **1)** We propose `Lumos-1`, a pure LLM-based unified architecture with discrete diffusion for AR video generation. **2)** We identify that although being useful, naive 3D RoPE exhibits imbalanced frequency spectra. Therefore, we propose MM-RoPE with more distributed frequency allocations and principled scaling, to achieve better dependency modeling and modality balancing. **3)** We identify the loss imbalance issue during mask prediction training, and propose AR-DF to enforce training with temporal modeling and introduce inference-time masks to avoid quality degradation. **4)** We validate the efficacy of the above designs through extensive experiments. `Lumos-1` achieves competitive results on T2I, I2V and T2V tasks, while using limited training computing, data and a discrete tokenizer.

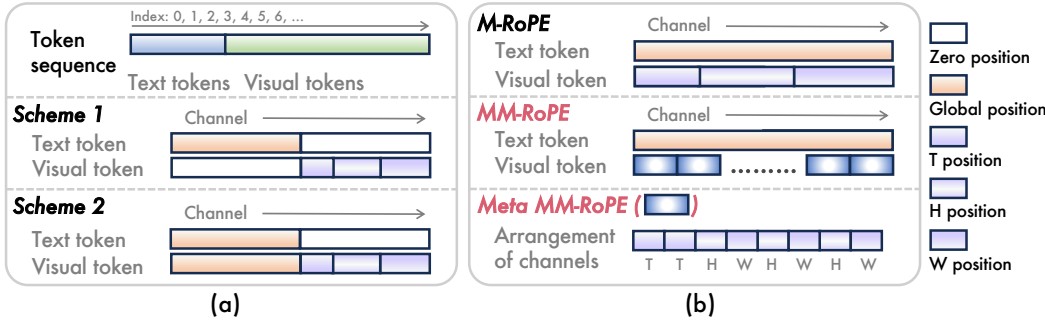

Figure 1: (a) **Initial exploration of 3D RoPE in autoregressive video generation**. The sequence is comprised of text tokens and visual tokens (default arrangement in this paper), with global indices starting from 0. Two schemes are exemplified using one text token and one visual token. In Scheme 1 and 2, temporal, height and width positions start from 0. (b) **Details of MM-RoPE** compared to M-RoPE. The figure illustrates the distributed channel allocation of MM-RoPE. Temporal, height and width positions are indexed after text tokens.

## 2 RELATED WORK

**Autoregressive video generation.** Autoregressive video synthesis follows a trajectory similar to images (from a dedicated architecture to a unified one like LLMs), but the substantial computational overhead encouraged researchers to explore different granularities of autoregression. At the *macro* level, methods such as CausVid Yin et al. (2024), Pyramidal Flow Jin et al. (2024), and MAGI-1 Sand-AI (2025) generate video clips by recursively invoking diffusion models. A second line of work adopts a *hybrid AR-diffusion* strategy that couples a pre-trained diffusion backbone with an external autoregressive planner, as exemplified by ARLON Li et al. (2024b) and Mardini Liu et al. (2024c). Our focus, however, is on *micro* AR approaches Gu et al. (2025); Zhou et al. (2025); Yu et al. (2023a;b); Villegas et al. (2022); Kondratyuk et al. (2023); Wang et al. (2024c); Deng et al. (2024); Wang et al. (2024d), which treat the entire spatiotemporal token sequence as a single context and generate videos end-to-end with one autoregressive transformer (*e.g.*, Phenaki Villegas et al. (2022), VideoPoet Kondratyuk et al. (2023), Loong Wang et al. (2024d), EMU3 Wang et al. (2024c), NOVA Deng et al. (2024)). Building on this paradigm, we systematically study the use of RoPE, and further introduce AR-DF, which ensures efficiency and is compatible with intra-frame bidirectionality and inter-frame temporal causality, enabling effective training and inference with LLM architectures.

**RoPE for vision-language data.** RoPE was first proven effective for LLMs Su et al. (2024); Touvron et al. (2023a); Huang et al. (2025a) and later adopted in DiTs Peebles & Xie (2023) to inject spatial priors. However, its potential for AR video generation is heavily underexplored. Related works, including M-RoPE Wang et al. (2024b), U-RoPE Tang et al., RoPE-2D Agrawal et al. (2024), VoPE Chen et al. (2024b), TAD-RoPE Gao et al. (2024), IL-RoPE Liao et al. (2025), VideoRoPE Wei et al. (2025) and HoPE Li et al. (2025a), incorporated structure priors into RoPE (*i.e.*, spatiotemporal information or its subsets) or improve the frequency allocation of RoPE for better vision-language understanding. However, they bear drawbacks: **1)** some of them are not compatible with the original text RoPE; **2)** their frequency allocations remain suboptimal for local-global dependency modeling; **3)** their scaling strategy remains suboptimal for spatiotemporal data. We propose MM-RoPE, a distributed and scaled RoPE technique that improves AR video synthesis while remaining plug-and-play for unified models.

## 3 LUMOS−1

### 3.1 SPATIOTEMPORAL CORRELATION INJECTION VIA MM-ROPE

**Preliminaries of 3D RoPE.** RoPE Su et al. (2024) aims to encode the absolute position with a rotation matrix while incorporating the explicit relative position dependency in the attention mechanism. If we denote $f_q(\boldsymbol{x}_m, m)$ and $f_k(\boldsymbol{x}_n, n)$ as query and key features encoding positions $m$ and $n$, the attention calculation in RoPE can be rewritten as:

$$f_q(\boldsymbol{x}_m, m)^{\mathrm{T}} f_k(\boldsymbol{x}_n, n) = \boldsymbol{x}_m^{\mathrm{T}} \boldsymbol{W}_q^{\mathrm{T}} \boldsymbol{R}_{\Theta,\tau}^d \boldsymbol{W}_k \boldsymbol{x}_n, \quad \tau = n - m \tag{1}$$

where $\boldsymbol{W}_{q,k}$ is the projection matrix; $\boldsymbol{R}_{\Theta,\tau}^d$ is the rotary matrix with pre-defined parameters $\Theta = \{\theta_i = \beta^{-2(i-1)/d}, i = [1, 2, ..., d/2]\}$ ($d$ is the feature dimension and $\beta$ is the base frequency). We

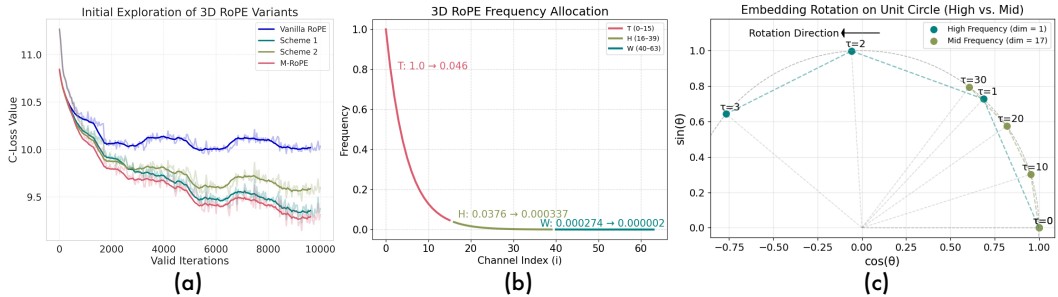

Figure 2: (a) **Validation loss curve of a toy experiment exploring the necessity of 3D RoPE** using the 0.5B model; (b) **Frequency allocation in the vanilla 3D RoPE**; (c) **Comparison of rotary speed** of the first dimension for temporal and height dimensions (high and middle frequencies) in the vanilla 3D RoPE.

formulate $\boldsymbol{R}_{\Theta,\tau}^{d}$ using a base rotary matrix $R_{\theta,\tau}$, with $\theta$ as the frequency and $\tau$ as the relative position:

$$\boldsymbol{R}_{\Theta,\tau}^{d} = \begin{bmatrix} R_{\theta_1,\tau} & 0 & \cdots & 0 \\ 0 & R_{\theta_2,\tau} & \cdots & 0 \\ \vdots & \vdots & \ddots & 0 \\ 0 & 0 & \cdots & R_{\theta_{d/2},\tau} \end{bmatrix}, \quad R_{\theta,\tau} = \begin{bmatrix} \cos\tau\theta & -\sin\tau\theta \\ \sin\tau\theta & \cos\tau\theta \end{bmatrix} \quad (2)$$

However, applying the original RoPE to modeling visual data remains suboptimal considering the spatiotemporal nature of visual tokens. Diffusion models Ho et al. (2022) improved upon this by proposing 3D RoPE that jointly injects spatiotemporal latent coordinates during attention calculation Hong et al. (2022); Kong et al. (2024). If we slightly abuse the annotation by denoting $\boldsymbol{x}_m^{\mathrm{T}}\boldsymbol{W}_q^{\mathrm{T}}$ and $\boldsymbol{W}_k\boldsymbol{x}_n$ as $\boldsymbol{X}_m^{\mathrm{T}}$ and $\boldsymbol{X}_n$, we can write the attention calculation based on 3D RoPE as:

$$\boldsymbol{X}_{m,t_s:t_e}^{\mathrm{T}} \begin{bmatrix} R_{\theta_{t_s+1},\tau_t} & \cdots & 0 \\ \vdots & \ddots & \vdots \\ 0 & \cdots & R_{\theta_{t_e},\tau_t} \end{bmatrix} \boldsymbol{X}_{n,t_s:t_e} + \boldsymbol{X}_{m,h_s:h_e}^{\mathrm{T}} \begin{bmatrix} R_{\theta_{h_s+1},\tau_h} & \cdots & 0 \\ \vdots & \ddots & \vdots \\ 0 & \cdots & R_{\theta_{h_e},\tau_h} \end{bmatrix} \boldsymbol{X}_{n,h_s:h_e} + \boldsymbol{X}_{m,w_s:w_e}^{\mathrm{T}} \begin{bmatrix} R_{\theta_{w_s+1},\tau_w} & \cdots & 0 \\ \vdots & \ddots & \vdots \\ 0 & \cdots & R_{\theta_{w_e},\tau_w} \end{bmatrix} \boldsymbol{X}_{n,w_s:w_e}$$
$$(3)$$

where $\{t_s, t_e\} = \{0, \frac{2}{16}d\}$, $\{h_s, h_e\} = \{\frac{2}{16}d, \frac{5}{16}d\}$ and $\{w_s, w_e\} = \{\frac{5}{16}d, \frac{1}{2}d\}$ denote the start and end dimension index for encoding temporal, height and width relative position; $\boldsymbol{X}_{m,t_s:t_e}^{\mathrm{T}}$ denotes the submatrix extracted from $\boldsymbol{X}_m^{\mathrm{T}}$ using row indices $[2t_s, 2t_e)$; other matrices are similarly defined.

**Initial exploration of incorporating 3D RoPE.** Our preliminary exploration starts with the introduction of 3D RoPE to AR video generation. We follow previous works Fan et al. (2024); Li et al. (2024a) to *utilize the validation loss to observe the effectiveness* due to its strong correlation with evaluation metrics. By default, we use the cross-entropy loss (C-Loss), following standard LLM training. We compare the vanilla LLM RoPE with three schemes, as shown in Fig. 1: **1)** *Scheme 1*, which allocates the first $1/2$ channels to encode the global position (*i.e.*, the index in the global sequence) and the second $1/2$ channels to encode the temporal, height and width positions with a ratio of $2 : 3 : 3$. For text tokens, we only use the first half channels to encode the global positions to ensure the language modeling capability, while for visual tokens, we only use the second half to encode 3D positions; **2)** *Scheme 2*, which extends upon scheme 1 by leveraging the first half channels of the visual tokens to encode the global positions. **3)** *M-RoPE* Wang et al. (2024b), which uses all channels of the visual tokens to encode the 3D positions. From Fig. 2(a), we can observe that: **1)** The incorporation of the spatiotemporal correlation in RoPE can significantly improve the performance of the model fitting data by comparing vanilla RoPE and Scheme 1. **2)** The injection of the raster-scan order position information to visual tokens (*i.e.*, the global position in Scheme 2 of Fig. 1(a)) can degrade the performance. **3)** A comprehensive channel utilization (M-RoPE) is better than partial channel utilization (Scheme 1). Therefore, it is promising to inject such priors in `Lumos-1`.

**Peering into 3D RoPE and identifying its limitations.** Although 3D RoPE proves effective in practice, its design remains suboptimal. In Fig. 2(b), we visualize how the frequencies are allocated to model temporal, height, and width dimensions. We observe that the *temporal channels dominate large frequency spectrum*, whereas the height and width channels are relegated to near-zero frequencies. For the sine function, the relative positions $\tau$ (when $\tau >= 0$) should not exceed one period to avoid ambiguity since radians beyond $2\pi$ start repeating patterns in the function. Beyond this range, the model cannot distinguish fine-grained positional differences since rotary embeddings of this channel lose uniqueness. For the low-indexed channels, the embeddings can rotate significantly faster than

**Algorithm 1** AR-DF Training Procedure

**Require:** video dataset $\mathcal{D}$, training mask ratio $\rho_{tra}$, generative model $G_\phi$, number of latent frames $T$
1: **for each** video $\boldsymbol{X} \in \mathcal{D}$ **do**
2:    $\boldsymbol{X}_p, \boldsymbol{X}_v \leftarrow$ tokenize text, frames
3:    **Sample a mask pattern for all frames**:
4:      $\rho \sim \text{Uniform}([\rho_{tra}, 1])$;
5:      $\boldsymbol{M} \sim \text{Bernoulli}(1 - \rho)$
6:    **for** $t = 1 \rightarrow T$ **do** ▷ Apply $\boldsymbol{M}$ to all frames
7:      $\widetilde{\boldsymbol{X}}_v^{(t)} = \boldsymbol{M} \odot \boldsymbol{X}_v^{(t)} + (1 - \boldsymbol{M}) \odot$ [MASK]
8:    **end for**
9:    Form $\widetilde{\boldsymbol{X}} = \{\boldsymbol{X}_p, \widetilde{\boldsymbol{X}}_v^{(1)}, \ldots, \widetilde{\boldsymbol{X}}_v^{(T)}\}$
10:    Generate attention mask $AttnMask$ for $\widetilde{\boldsymbol{X}}$
11:    $\widehat{\boldsymbol{X}} \leftarrow G_\phi(\widetilde{\boldsymbol{X}}, AttnMask)$
12:    Compute loss $\mathcal{L}(\widehat{\boldsymbol{X}}, \boldsymbol{X}, \boldsymbol{M})$
13:    Backprop & update $\phi$
14: **end for**

**Algorithm 2** AR-DF Inference Procedure

**Require:** text prompt $\boldsymbol{X}_p$, trained model $G_\phi$, inference mask ratio $\rho_{inf}$, number of latent frames $T$, number of generation steps $N_{steps}$, number of tokens in a latent frame $N_f$, KV cache $\mathcal{C} = \varnothing$, generated latent list $\mathcal{V} = \varnothing$
1: **Initialize text cache**:
2:    Generate text causal mask $AttnMask^{(p)}$ for $\boldsymbol{X}_p$
3:    $\mathcal{C} \leftarrow G_\phi(\boldsymbol{X}_p, AttnMask^{(p)})$ ▷ Store cache for the prompt
4: **Sample cache mask**:
5:    $\boldsymbol{M}_{inf} \sim \text{Bernoulli}(1 - \rho_{inf})$
6: **for** $t = 1 \rightarrow T$ **do** ▷ Reused for all frames
7:    Initialize all tokens in $\boldsymbol{X}_v^{(t)}$ as [MASK]
8:    Generate temporal causal mask $AttnMask^{(t)}$ for $\{\boldsymbol{X}_p, \boldsymbol{X}_v^{(t)}\}$
9:    Generate $t$-th frame $\boldsymbol{X}_v^{(t)}$ and append it to $\mathcal{V}$
10:    **Cache partial observation of the generated frame**:
11:      $\widetilde{\boldsymbol{X}}_v^{(t)} = \boldsymbol{M}_{inf} \odot \boldsymbol{X}_v^{(t)} + (1 - \boldsymbol{M}_{inf}) \odot$ [MASK]
12:      $\mathcal{C} \leftarrow G_\phi(\widetilde{\boldsymbol{X}}_v^{(t)}, AttnMask^{(t)}, \mathcal{C})$ ▷ Store cache for the $t$-th frame
13: **end for**

Figure 3: Left: **Training algorithm with AR-DF**. For simplicity, this only includes training on videos. Right: **Inference algorithm with AR-DF.** A more detailed version for inference is included in the Appendix.

high-indexed channels (as shown in Fig. 2(c)), leading to accelerated aliasing and loss of embedding uniqueness. For the high-indexed channels, the embeddings rotate so slowly that they lack sufficient resolution to model subtle local changes. Finally, while height and width are symmetrically important, they occupy disproportionately small and different segments of the overall frequency spectrum, reducing their capacity to capture spatial details effectively.

**MM-RoPE: a distributed and scaled 3D RoPE mechanism.** To elegantly solve the aforementioned limitation, we propose MM-RoPE, a distributed 3D RoPE mechanism. Compared with M-RoPE Wang et al. (2024b) that is widely adopted in vision-language models, one core idea of MM-RoPE is to encode relative positions across comprehensive frequency spectra for all 3D information. As illustrated in Fig. 1(b), the RoPE for text tokens in MM-RoPE follows the standard design of an LLM, whereas the RoPE for visual tokens is comprised of multiple meta MM-RoPE components. Within each meta MM-RoPE, we keep the ratio of 3D information identical to 3D RoPE (*i.e.*, $2 : 3 : 3$), while minimizing the overall dimensionality to maintain a more distributed design. Concretely, we first allocate channels for temporal modeling, then symmetrically interleave height and width channels to model spatial information. We can formulate the attention calculation of the first meta MM-RoPE as:

$$
\begin{aligned}
&\boldsymbol{X}_{m,0:2}^{\mathrm{T}} \begin{bmatrix} R_{\theta_1,\tau_t} & 0 \\ 0 & R_{\theta_2,\tau_t} \end{bmatrix} \boldsymbol{X}_{n,0:2} + \boldsymbol{X}_{m,2:4}^{\mathrm{T}} \begin{bmatrix} R_{\theta_3,\tau_h} & 0 \\ 0 & R_{\theta_4,\tau_w} \end{bmatrix} \boldsymbol{X}_{n,2:4} + \\
&\boldsymbol{X}_{m,4:6}^{\mathrm{T}} \begin{bmatrix} R_{\theta_5,\tau_h} & 0 \\ 0 & R_{\theta_6,\tau_w} \end{bmatrix} \boldsymbol{X}_{n,4:6} + \boldsymbol{X}_{m,6:8}^{\mathrm{T}} \begin{bmatrix} R_{\theta_7,\tau_h} & 0 \\ 0 & R_{\theta_8,\tau_w} \end{bmatrix} \boldsymbol{X}_{n,6:8}
\end{aligned}
\tag{4}
$$

where each meta MM-RoPE component comprises 16 channels; additional components are defined analogously and collectively form the RoPE strategy for visual tokens.

Moreover, for a model that jointly processes text and visual tokens, the interplay between two modalities is significant to ensure the vision-language alignment. However, the range of positions for representing texts or visual data tends to differ. Contemporary visual generation systems are typically trained with extremely long and descriptive captions Betker et al. (2023); Team (2025), despite the low latent resolution of visual data (*e.g.*, a video of resolution $448 \times 256 \times 25$ becomes $56 \times 32 \times 7$ after a $8 \times 8 \times 4$ compression).

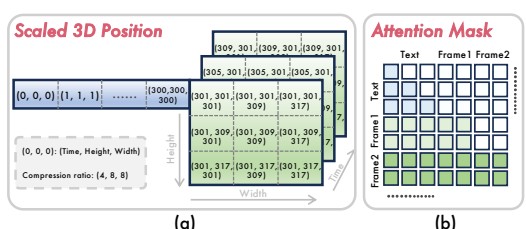

Figure 4: (a) **Details of scaled 3D positions in MM-RoPE.** (b) **Temporal causal mask used in `Lumos-1`.**

To balance the two modalities, we propose scaling the 3D positions to ensure a balanced learning. Specifically, we empirically scale the latent 3D positions to the RGB space by multiplying the compression ratio, as shown in Fig. 4(a). This simple scaling operation, from another perspective, improves the RoPE resolution for visual tokens by slightly accelerating the rotation speed. Experiments in the experiment section demonstrate its efficacy, thereby showing the importance of balancing the two modalities from the RoPE perspective.

We acknowledge, however, that given the autoregressive generation nature of videos, this scaling may not be the optimal solution. More advanced and sophisticated solutions are left for future work.

## 3.2 Autoregressive Discrete Diffusion Forcing

The most naive generation paradigm (*i.e.*, next-token prediction) suffers from incompatibility with the nature of videos and low generation efficiency, rendering it impractical for autoregressive visual generation. In this work, we resort to discrete diffusion Xie et al. (2024); Swerdlow et al. (2025) to generate visual content, together with spatially bidirectional and temporally causal token dependency. However, due to the autoregressive nature of Lumos-1, naive random masking (*i.e.*, a globally random mask) or temporally independent masking (*i.e.*, diffusion forcing Chen et al. (2024a); Song et al. (2025)) both lead to significant loss imbalance, *i.e.*, visual tokens in the later frames tend to have much lower loss. Since the task difficulty of predicting frames with ample history frame context is considerably easier than predicting the first image given a text prompt, the model would lean towards optimizing simpler tasks, leading to a degradation of temporal learning.

**Training scheme.** To resolve this issue, we build upon the basic nature of videos–spatial information redundancy. The core issue of imbalanced loss during training stems from spatial information leakage. It is worth noting that the original video diffusion transformer Song et al. (2025) that adopts diffusion forcing does not encounter this issue due to its usage of bidirectional dependency. Facing this challenge, we introduce Autoregressive Discrete Diffusion Forcing (AR-DF), which adopts temporal tube masking during the training of AR video generation. Concretely, for every video, we randomly generate a mask pattern for the first frame and then apply this mask pattern repeatedly to later frames in this video. If we denote the multi-modal token sequence $X$ composed of text tokens $X_p$ and visual tokens $X_v$ and sample a mask ratio $\rho$, the training masking strategy in AR-DF can be formulated as:

$$M_i \sim \text{Bernoulli}(1 - \rho) \quad \text{for } i = 1, \ldots, N_f \tag{5}$$

$$\widetilde{X}_v^{(t)} = M \odot X_v^{(t)} + (1 - M) \odot [\text{MASK}], \quad t = 1, \ldots, T \tag{6}$$

$$\widetilde{X} = \left\{ X_p, \ \widetilde{X}_v^{(1)}, \ \widetilde{X}_v^{(2)}, \ \ldots, \ \widetilde{X}_v^{(T)} \right\} \tag{7}$$

where $N_f$ and $T$ denote the number of tokens in a latent frame and the number of latent frames in $X_v$; $X_v^{(t)}$ denotes the visual tokens of the $t$-th frame; $\widetilde{X}$ denotes the masked multi-modal token sequence prepared for training; $\odot$ indicates Hadamard multiplication; $M$ denotes the mask pattern; $[\text{MASK}]$ denotes the mask token. After the preparation of the token sequence, it is fed into the model for processing. To ensure consistency with native LLMs and temporal causality in videos, we adopt a temporal causal mask $AttnMask$ for attention processing, as shown in Fig. 4(b). To train the model, we use cross-entropy loss and compute the loss only on masked tokens, denoted as $\mathcal{L}(\widehat{X}, X, M)$ and $\widehat{X}$ is the token sequence after model processing. The algorithm is formalized in Algorithm 1.

**Inference scheme.** After training with AR-DF, the most naive inference scheme (*i.e.*, autoregressively generating video frames) would result in significant frame quality and motion degradation. We observe that this is caused by inconsistent inference with training. During training, later frames consistently have partial observation of history frames, while the inference stage does not align with this observation pattern. Therefore, given a caption, we first generate the first frame by running multiple steps and then randomly replace a pre-defined ratio $\rho_{inf}$ of tokens with the [MASK] token for the generated image. We infer the model with this partially observed image and cache the Keys and Values of this image for swift inference. This process is repeated until the entire video is generated. The algorithm is formalized in Algorithm 2.

## 3.3 Implementation

**Architecture.** Lumos-1's architecture follows Llama Touvron et al. (2023b); Grattafiori et al. (2024). To stabilize training, we integrate QK-Norm following Chameleon Team (2024). We have models of three scales (0.5B, 1B and 3B), whose architectural details are placed in the Appendix.

**Tokenizers.** We adopt the discrete Cosmos Tokenizer Agarwal et al. (2025) that achieves spatiotemporal compression rates of $8 \times 8 \times 4$. For text tokens, we retain Chameleon's text tokenizer. Lumos-1's total codebook size is 129,536, partitioned into 65,536 text tokens and 64,000 visual tokens.

**Stage-wise training.** Due to the autoregressive nature of Lumos-1, the training of video generation can be categorized into training two capabilities: **1)** text-to-image and **2)** image/images to video. Although the incorporation of AR-DF training substantially ameliorates the imbalanced learning

Table 2: **Performance comparison on GenEval.** The "#Params" column shows the parameter counts of the generative model and the language encoder for a fair comparison to unified models following Zhou et al. (2024). **Gen Rep** denotes the generation representation type: continuous or discrete. * denotes results without inference-time scaling. † denotes supervised fine-tuning on small-scale data.

| Model | #Params | #Images | Gen Rep | Overall↑ | Single Obj. | Two Obj. | Counting | Colors | Position | Attr. Bind |
|---|---|---|---|---|---|---|---|---|---|---|
| **Diffusion models** | | | | | | | | | | |
| SD v1.5 Rombach et al. (2022) | 0.9B + 0.1B | 2B | Continuous | 0.43 | 0.97 | 0.38 | 0.35 | 0.76 | 0.04 | 0.06 |
| SD-XL Podell et al. (2024) | 2.6B + 0.8B | – | Continuous | 0.55 | 0.98 | 0.74 | 0.39 | 0.85 | 0.15 | 0.23 |
| SD 3 Esser et al. (2024) | 8.2B + 2.8B | – | Continuous | 0.68 | 0.98 | 0.84 | **0.66** | 0.74 | 0.40 | 0.43 |
| DALL-E 2 Ramesh et al. (2022) | 4.2B + 1.0B | 650M | Continuous | 0.52 | 0.94 | 0.66 | 0.49 | 0.77 | 0.10 | 0.19 |
| SANA 1.5 Xie et al. (2024) | 1.6B + 2.6B | 50M | Continuous | 0.66 | – | – | – | – | – | – |
| FLUX Labs (2024) | 12B + 2.5B | – | Continuous | 0.67 | 0.99 | 0.85 | 0.75 | 0.77 | 0.22 | 0.42 |
| Meissonic Bai et al. | 1B + 0.4B | 210M | Discrete | 0.54 | 0.99 | 0.66 | 0.42 | 0.86 | 0.10 | 0.22 |
| Muddit Shi et al. (2025) | 1B + 0.4B | 50M | Discrete | 0.61 | 0.98 | 0.72 | 0.54 | 0.82 | 0.19 | 0.41 |
| UniDisc Swerdlow et al. (2025) | 1.4B | 280M | Discrete | 0.42 | 0.92 | 0.47 | 0.15 | 0.67 | 0.13 | 0.19 |
| D-DiT Li et al. (2025b) | 2B | – | Discrete | 0.65 | 0.97 | 0.80 | 0.54 | 0.76 | 0.32 | 0.50 |
| MMaDA Yang et al. (2025) | 8B | – | Discrete | 0.63 | 0.99 | 0.76 | 0.61 | 0.84 | 0.20 | 0.37 |
| **Autoregressive models** | | | | | | | | | | |
| SEED-X Ge et al. (2024) | 17B | – | Continuous | 0.49 | 0.97 | 0.58 | 0.26 | 0.80 | 0.19 | 0.14 |
| Transfusion Zhou et al. (2024) | 7.3B | 3.5B | Continuous | 0.63 | – | – | – | – | – | – |
| Fluid Fan et al. (2024) | 10.5B + 4.7B | 680M | Continuous | 0.69 | 0.96 | 0.83 | 0.63 | 0.80 | 0.39 | 0.51 |
| Show-o2 Xie et al. (2025) | 7B | 66M | Continuous | 0.76 | **1.00** | 0.87 | 0.58 | **0.92** | 0.52 | 0.62 |
| LlamaGen Sun et al. (2024) | 0.8B + 2.9B | 60M | Discrete | 0.32 | 0.71 | 0.34 | 0.21 | 0.58 | 0.07 | 0.04 |
| Show-o Xie et al. (2024) | 1.3B | 2B | Discrete | 0.68 | 0.98 | 0.80 | 0.66 | 0.84 | 0.31 | 0.50 |
| Chameleon Team (2024) | 34B | 1.4B | Discrete | 0.39 | – | – | – | – | – | – |
| Lumina-mGPT Liu et al. (2024b) | 7B | 1.4B | Discrete | 0.56 | – | 0.77 | 0.27 | – | – | 0.32 |
| Lumina-mGPT 2.0* Xin et al. (2025) | 7B | – | Discrete | 0.73 | **1.00** | **0.87** | 0.49 | 0.85 | 0.52 | 0.62 |
| EMU3 Zhou et al. (2024) | 8B | – | Discrete | 0.66 | 0.99 | 0.81 | 0.42 | 0.80 | 0.49 | 0.45 |
| Lumos-1 (352 × 352) | 1.5B | 60M | Discrete | 0.601 | 0.959 | 0.732 | 0.375 | 0.774 | 0.365 | 0.400 |
| Lumos-1 (352 × 352) | 3.6B | 60M | Discrete | 0.664 | 0.953 | 0.806 | 0.463 | 0.806 | 0.483 | 0.475 |
| Lumos-1 (512 × 512)† | 1.5B | 60M | Discrete | 0.725 | 0.984 | **0.869** | 0.519 | 0.862 | 0.558 | 0.558 |
| Lumos-1 (512 × 512)† | 3.6B | 60M | Discrete | **0.791** | 0.991 | 0.866 | **0.678** | 0.840 | **0.683** | **0.688** |

Table 3: **Performance comparison on VBench-I2V benchmark.** We list partial metrics due to space limits.

| Model | #Params | #Videos | Gen Rep | Total↑ | I2V Score | Quality Score | I2V Sub. | I2V Back. | Sub. Cons. | Back. Cons. | Img. Quality |
|---|---|---|---|---|---|---|---|---|---|---|---|---|
| **Diffusion models** | | | | | | | | | | | | |
| VideoCrafter-I2V Chen et al. (2023a) | 2.6B | 20M | Continuous | 82.57 | 86.31 | 78.84 | 91.17 | 91.31 | 97.86 | 98.79 | 71.68 |
| ConsistI2V Ren et al. (2024b) | 1.3B + 0.3B | 10M | Continuous | 84.07 | 91.91 | 76.22 | 95.82 | 95.95 | 95.27 | 98.28 | 66.92 |
| SEINE Chen et al. (2023b) | 0.9B + 0.1B | 10M | Continuous | 84.88 | 92.39 | 77.37 | 96.57 | 96.80 | 94.2 | 97.26 | 70.97 |
| I2VGen-XL Zhang et al. (2023) | 1.4B + 1.0B | 35M | Continuous | 85.28 | 92.11 | 78.44 | 96.48 | 96.83 | 94.18 | 97.09 | 69.14 |
| CogVideoX Yang et al. (2024b) | 5.6B + 4.8B | – | Continuous | 86.70 | 94.79 | 78.61 | 97.19 | 96.74 | 94.34 | 96.42 | 70.01 |
| **Autoregressive models** | | | | | | | | | | | | |
| COSMOS Sun et al. (2024) | 5B + 11B | 100M | Continuous | 84.16 | 92.51 | 75.81 | 95.99 | 97.36 | 97.12 | 96.59 | 59.90 |
| VideoMAR Hu Yu (2025) | 1.4B + 1.5B | 0.5M | Continuous | 84.82 | 94.02 | 75.6 | 97.85 | 98.38 | 97.13 | 97.20 | 62.34 |
| Lumos-1 (672 × 384 × 25) | 1.5B | 10M | Discrete | 84.16 | 91.80 | 76.53 | 96.06 | 96.58 | 95.90 | 96.25 | 67.06 |
| Lumos-1 (672 × 384 × 25) | 3.6B | 10M | Discrete | 84.72 | 93.34 | 76.10 | 97.42 | 97.40 | 97.42 | 96.91 | 69.23 |

issue, we still observe that the later task is relatively easier. Therefore, a stage-wise training scheme is mandatory to ensure successful video generation training. Our training consists of three pre-training stages and a supervised fine-tuning (SFT) stage. The pre-training stages focus on learning the generation capabilities from a vast amount of data: **1)** text-to-image training at 256p, **2)** joint image-video training at 256p, and **3)** joint fine-tuning at 384p. The final SFT stage focuses on small-scale fine-tuning on data of high quality. We use around 100k images and 20k videos. This process teaches the model to more precisely follow instructions and adhere to specific aesthetic styles. We find that both models trained on 256p or 384p data can be boosted by simple SFT.

Details on *sequence formatting* and *GPU memory friendly implementation* are placed in the Appendix.

## 4 EXPERIMENTS

### 4.1 EXPERIMENTAL DETAILS

**Datasets.** To train Lumos-1, we curate a image dataset containing 60 million images and a video dataset containing 10 million videos. We preserve their original aspect ratios, with videos being clipped to 25 frames for training. To ensure fine-grained vision-language alignment Ramesh et al. (2022); Team (2025), the visual data is re-captioned using visual-language models Lin et al. (2024b) to obtain long and descriptive captions.

**Training, inference and model evaluation.** We train the model from scratch and the basic training hyper-parameters are placed in the Appendix. During AR-DF training, the training mask ratio $\rho_{tra}$ is set to 0.7 following Li et al. (2024a). During AR-DF inference, classier-free guidance (CFG) is utilized by default to enhance the generation quality. The inference mask ratio $\rho_{inf}$ is set to 0.7 by default. The number of steps to generate one latent frame is set to $N_{steps} = 50$ by default. When evaluating on GenEval Ghosh et al. (2023) and VBench Huang et al. (2024), the guidance scale is set to 16 if not otherwise specified. Since we train our model only on detailed and descriptive captions, it is mandatory for us to rewrite captions using Qwen 32B Yang et al. (2024a) when evaluating on

Table 4: **Performance comparison on VBench-T2V benchmark.** We list partial metrics due to space limits. † denotes supervised fine-tuning on small-scale data.

| Model | #Params | #Videos | Gen Rep | Total↑ | Quality | Semantic | Sub. Cons. | Back. Cons. | Img. Quality | Obj. Class | Color | Overall Cons. |
|---|---|---|---|---|---|---|---|---|---|---|---|---|
| **Diffusion models** | | | | | | | | | | | | |
| ModelScopeT2V Wang et al. (2023a) | 1.4B + 0.3B | 10M | Continuous | 75.75 | 78.05 | 66.54 | 89.87 | 95.29 | 58.57 | 82.25 | 81.72 | 25.67 |
| InstructVideo Yuan et al. (2024) | 1.4B + 0.3B | 10M | Continuous | 76.61 | 81.56 | 56.81 | 95.30 | 96.97 | 68.01 | 73.26 | 77.14 | 19.91 |
| LaVie Wang et al. (2023b) | 2.5B + 0.5B | 25M | Continuous | 77.08 | 78.78 | 70.31 | 91.41 | 97.47 | 61.90 | 91.82 | 86.39 | 26.41 |
| OpenSoraPlan V1.3 Lin et al. (2024a) | 2.7B + 13B | 70M | Continuous | 77.23 | 80.14 | 65.62 | 97.79 | 97.24 | 56.21 | 85.56 | 79.30 | 24.47 |
| LTX-Video HaCohen et al. (2024) | 1.9B + 4.8B | – | Continuous | 80.00 | 82.30 | 70.79 | 96.56 | 97.20 | 60.28 | 83.45 | 81.45 | 25.19 |
| CogVideoX Yang et al. (2024b) | 5.6B + 4.8B | – | Continuous | 81.91 | 83.05 | 77.33 | 96.45 | 96.71 | 63.33 | 85.07 | 83.03 | 27.65 |
| **Autoregressive models** | | | | | | | | | | | | |
| CogVideo Hong et al. (2022) | 9B | 5.4M | Continuous | 67.01 | 72.06 | 46.83 | 92.19 | 96.20 | 41.03 | 73.40 | 79.57 | 7.70 |
| NOVA Deng et al. (2024) | 0.6B + 2.8B | 20M | Continuous | 80.12 | 80.39 | 79.05 | – | – | – | 92.00 | – | – |
| EMU3 Wang et al. (2024c) | 8B | – | Discrete | 80.96 | 84.09 | 68.43 | 95.32 | 97.69 | – | 86.17 | – | – |
| Lumos-1 (672 × 384 × 25) | 1.5B | 10M | Discrete | 76.34 | 78.27 | 68.65 | 96.04 | 96.29 | 56.16 | 90.47 | 77.49 | 25.07 |
| Lumos-1 (672 × 384 × 25) | 3.6B | 10M | Discrete | 78.32 | 79.52 | 73.51 | 95.51 | 96.50 | 58.04 | 90.05 | 82.00 | 25.29 |
| Lumos-1 (672 × 384 × 25)† | 1.5B | 10M | Discrete | 78.17 | 79.60 | 72.45 | 95.36 | 96.08 | 57.21 | 93.39 | 77.13 | 25.55 |
| Lumos-1 (672 × 384 × 25)† | 3.6B | 10M | Discrete | 78.90 | 79.83 | 75.21 | 96.78 | 96.60 | 61.92 | 94.38 | 81.14 | 25.57 |

GenEval Ghosh et al. (2023). When evaluating on VBench Huang et al. (2024), we use its official long captions by default.

## 4.2 COMPARISON WITH OTHER METHODS ON VISUAL GENERATION

**Text-to-image generation.** We compare Lumos-1 with competitive image generation methods in Tab. 2. Compared with diffusion models, we observe that Lumos-1 outperforms competitive models (*e.g.*, SD-XL Podell et al. (2024) and FLUX Labs (2024)) using continuous or discrete tokenizers by a clear margin, even if Lumos-1 only uses a discrete tokenizer. Compared with autoregressive models, we observe that Lumos-1 outperforms EMU3 Wang et al. (2024c), Fluid Fan et al. (2024) and Lumina-mGPT series Xin et al. (2025); Liu et al. (2024b), while using substantially fewer training images and retaining a small model. Lumos-1 achieves superior results in terms of position and attribute binding, demonstrating excellent language understanding and vision-language alignment even without textual pre-training. We also observe that with a small-scale supervised fine-tuning using high-quality data, the performance can be largely boosted on GenEval, especially on challenging metrics like position and attribute binding.

**Image-to-video generation.** Thanks to the autoregressive nature of Lumos-1, we can perform image-to-video generation by specifying the first frame, although we did not specifically train on this task. Results are listed in Tab. 3. Lumos-1 outperforms the popular VideoCrafter-I2V model and is on par with the leading COSMOS-Video2World model, which uses substantially more data (100M>10M) and training resources (10000 H100s>48 H20s), demonstrating the promising performance of Lumos-1.

**Text-to-video generation.** We list the comparison in Tab. 4. Although Lumos-1 utilizes discrete tokenizers and does not rely on a heavy pre-trained text encoder, it can still be on par with diffusion models like Open-SoraPlan. Due to the autoregressive nature, the video quality can be ensured by first frame quality, enabling Lumos-1 to excel in object-centric metrics (object class and color).

Table 5: **Comparison with different training methods** for AR video generation. Results are measured on GenEval, Vbench-Overall Consistency (OC), and Vbench-Imaging Quality (IQ).

| Training Methods | GenEval | Vbench-OC | Vbench-IQ |
|---|---|---|---|
| Random masks | 0.593 | 0.232 | 0.424 |
| Linear decay loss | 0.588 | 0.228 | 0.410 |
| Step decay loss | 0.584 | 0.234 | 0.464 |
| Diffusion forcing masks | 0.590 | 0.241 | 0.540 |
| AR-DF | 0.591 | 0.249 | 0.559 |

## 4.3 ANALYSIS AND ABLATION STUDIES

**Qualitative visual comparison.** We compare Lumos-1 with popular video generation methods in Fig. 5. For T2V, the visual quality of our 384p videos does not trail 512p videos from LTX-Video. In the provided case, Lumos-1 generates better natural motion (water waves) and aligns with prompts better (skier in red and waves). For I2V, Lumos-1 handles multiple-object (multiple floating hot air balloons in example 1) and fine-grained motion (subtle ripples around the shoreline in example 3) substantially better than Stable Video Diffusion Blattmann et al. (2023) (SVD), which generates global camera movement only. In example 2, SVD produces significant blurs, whereas Lumos-1 animates subjects smoothly. ***More visualizations are placed in the Appendix.***

**Effectiveness of temporal-tube masks during AR-DF training.** In Fig. 6(a), we compare the frame-wise validation loss (frame 0, 3, 6) when using global random masks and temporal tube masks.

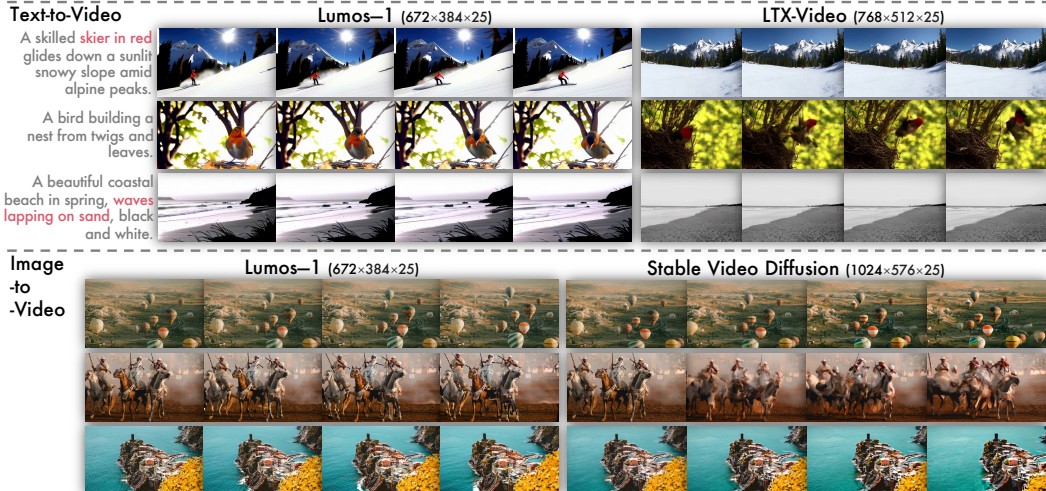

Figure 5: **Visual comparison** of `Lumos-1` with other methods on text/image-to-video tasks.

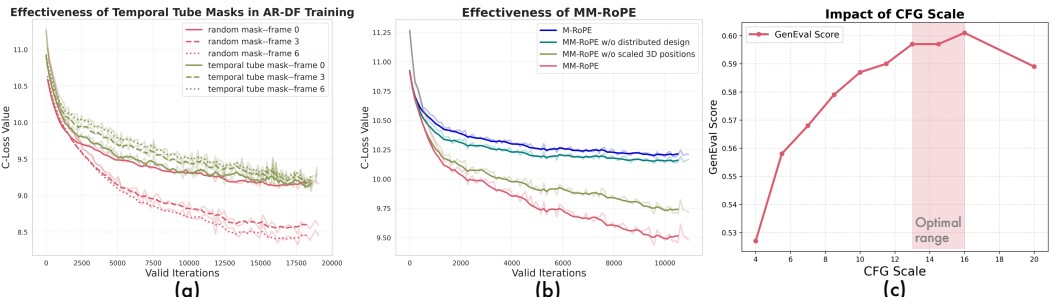

Figure 6: (a) **Effectiveness of temporal-tube masks** during AR-DF training (on videos) using the 0.5B model; (b) **Effectiveness of MM-RoPE**; (c) **Sensitivity analysis of CFG scale** on GenEval using 1B model.

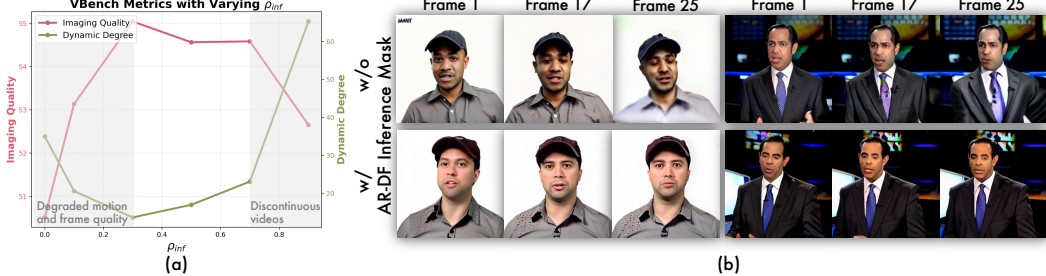

Figure 7: (a) **Selected VBench metrics with varying inference-time mask ratio** $\rho_{inf}$; (b) **Text-to-video visualization comparing the effect of inference-time masks**. (w/o: $\rho_{inf} = 0.0$, w/: $\rho_{inf} = 0.7$)

For random masks, the loss of frame 6 decreases immediately and becomes lower than that of earlier frames. This rapid fall indicates pronounced information leakage: the model can reconstruct a masked token by attending to unmasked tokens in neighbouring frames instead of modelling genuine temporal dynamics, rendering the task overly easy. For temporal tube masks, frame 6 is the hardest because it has the longest context to learn from, and all pixels in the same spatial locations are masked across the temporal axis, eliminating the learning shortcut. As iterations proceed, the gap between frames narrows and eventually levels out, demonstrating that the model is learning to propagate information through time rather than copying it.

**Effect of AR-DF inference masks and sensitivity to** $\rho_{inf}$. AR-DF requires the same partial-context masking at inference as during training; omitting these masks severely harms quality. In Fig. 7(b), we observe that the "w/o inference mask" setting pro-

Table 6: **Inference time analysis with different RoPEs and decoding strategies**. Results are reported on one H20 with batch size set to 1, $N_{steps} = 50$ and CFG used by default.

| | 1D RoPE (1B / 3B) | M-RoPE (1B / 3B) | MM-RoPE (1B / 3B) |
|---|---|---|---|
| Image (448 × 256) | 7.4s / 16.3s | 7.7s / 16.9s | 7.7s / 16.9s |
| Video (448 × 256 × 25) | 75.1s / 173.6s | 77.8s / 178.5s | 77.8s / 178.5s |
| | **Next-Token Pred** | **Mask Pred w/o KV Cache** | **Mask Pred w/ KV Cache** |
| Video (448 × 256 × 25) | 960.0s | 383.0s | 77.8s |

duces visible artifacts and flickering, whereas using masks preserves coherence. In Fig. 7(a), we select two VBench metrics (imaging quality and dynamic degree) to assess the impact of $\rho_{inf}$. A broad plateau between 0.3 and 0.7 yields smooth, visually pleasing videos. When $\rho_{inf} < 0.3$, insufficient context causes motion and per-frame quality degradation, thus inflating dynamic degree values. When $\rho_{inf} > 0.7$, aggressive masking disrupts temporal continuity, driving up dynamic degree values. We set $\rho_{inf}$ to 0.7 to ensure obvious motion.

**Training strategy comparison.** Temporal tube masking is a key in AR-DF training. In Tab. 5, we compare it with other methods, including global random masks, loss reweighting (step decay loss Wang et al. (2024d) and its modified version, linear decay loss), and diffusion forcing masks, by performing joint training on `Lumos-1` 1B using 8 H20 for 10k steps. We observe that loss reweighting cannot solve the information leakage issue. Diffusion forcing can ameliorate this, but temporal tube masking in AR-DF is particularly suited for videos, thus surpassing all other methods.

**Efficacy of MM-RoPE.** Fig. 6(b) plots the validation loss of the 0.5B model under four RoPE settings. Note that M-RoPE means that both designs are removed. We can observe that MM-RoPE consistently converges faster and settles at the lowest loss, confirming the benefit of modelling fine-grained spatiotemporal information. Although ablating either component raises the loss, dropping the distributed design hurts more than dropping the design of scaled position, indicating that comprehensive frequency allocation is the dominant factor. Removing both enhancements gives the slowest convergence and the highest plateau, showing that the two mechanisms are complementary for effective video generation. We quantitatively ablate on the MM-RoPE design in Tab. 7 by performing joint training on `Lumos-1` 1B using 8 H20 for 10k steps. We can find that the distributed design contributes significantly to the boost and speeds up the model's convergence. By varying the scaling factors, we confirm that (4,8,8) is a suitable and well-justified choice for visual generation.

Table 7: **Ablation study on MM-RoPE.**

| Distributed | Scaled 3D | GenEval | Vbench-OC | Vbench-IQ |
|---|---|---|---|---|
|  |  | 0.310 | 0.122 | 0.350 |
|  | ✓ | 0.414 | 0.211 | 0.451 |
| ✓ |  | 0.566 | 0.245 | 0.535 |
| ✓ | ✓ | 0.591 | 0.249 | 0.559 |

| Scaling Factors | | GenEval | Vbench-OC | Vbench-IQ |
|---|---|---|---|---|
| (1, 1, 1) | | 0.566 | 0.245 | 0.535 |
| (2, 4, 4) | | 0.583 | 0.248 | 0.545 |
| (4, 8, 8) | | 0.591 | 0.249 | 0.559 |
| (8, 16, 16) | | 0.593 | 0.250 | 0.553 |

**Inference time analysis.** In Tab. 6, we compare the inference speed using the vanilla 1D RoPE, M-RoPE and MM-RoPE. We can observe that: **1)** Compared with 1D RoPE, the incorporation of 3D priors introduces only 3.5%-4.1% inference latency. **2)** Compared to M-RoPE, MM-RoPE introduces no additional latency. We also compare the speed using different decoding strategies. We observe a clear efficiency boost compared with vanilla next-token prediction by using discrete diffusion (mask prediction) and KV cache.

**Comparison with different RoPE designs.** To validate the efficacy of MM-RoPE, we benchmark it against M-RoPE Wang et al. (2024b), VideoRoPE Wei et al. (2025), U-RoPE Tang et al., IL-RoPE Liao et al. (2025) and HoPE Li et al. (2025a). As shown in Tab. 8, MM-RoPE consistently outperforms them. This superiority stems from its unique design, which holistically incorporates a comprehensive frequency allocation strategy and a strategic scaling for modality alignment and resolution enhancement. A detailed breakdown of each method and a full analysis are provided in Appendix E.2.

Table 8: **Performance comparison with other types of RoPE** on GenEval and VBench.

| RoPE Type | GenEval | Vbench-OC | Vbench-IQ |
|---|---|---|---|
| M-RoPE | 0.310 | 0.122 | 0.350 |
| U-RoPE | 0.402 | 0.165 | 0.423 |
| IL-RoPE | 0.541 | 0.225 | 0.513 |
| VideoRoPE | 0.569 | 0.243 | 0.540 |
| HoPE | 0.570 | 0.246 | 0.545 |
| MM-RoPE | 0.591 | 0.249 | 0.559 |

**Sensitivity analysis of CFG scale.** We study the impact of the guidance scale on GenEval using 1B model in Fig. 6(c). We find that scale values from 13 to 16 (default value) lead to decent results.

## 5 CONCLUSION

In this paper, we introduce `Lumos-1`, which utilizes the LLM architecture for AR video generation. We propose MM-RoPE for better spatiotemporal dynamics modeling and propose AR-DF for effective training and inference considering intra-frame bidirectionality and inter-frame temporal causality. We anticipate that `Lumos-1` represents a significant step toward building a foundational unified model.

## ACKNOWLEDGEMENT

Hangjie Yuan was supported in part by the Postdoctoral Science Preferential Funding of Zhejiang Province, China (ZJ2025005).

## ETHICS STATEMENT

Our work introduces `Lumos-1`, an autoregressive model for text-to-video generation. Similar to other generative models, this technology carries societal risks if misused. The primary concerns include the potential for creating deceptive or misleading content for misinformation campaigns, the generation of harmful or disturbing visuals, and the potential amplification of societal biases present in the training data. To address these, we have taken several steps and advocate for further safeguards, as outlined in Appendix F.3. We firmly state that `Lumos-1` is currently a research-oriented project. We recommend that any future public release of this technology must be preceded by rigorous safety measures, also detailed in Appendix F.3.

## REPRODUCIBILITY STATEMENT

We are committed to ensuring the reproducibility of our work. All essential details for reproducing our results are provided within the paper and the appendix. The details of architectures and their training details are presented in and Sec. 3.3, Sec. 4.1 and Appendix D. Our technical contribution, MM-RoPE and AR-DF, are described with comprehensive details in Sec. 3. The evaluation protocols are detailed in Sec. 4.1. We will ensure the release of our source code, pre-trained model weights, and evaluation scripts upon publication of this work. We can also provide any source code if any reviewer asks for a detailed implementation.

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

APPENDIX

In this Appendix, we provide additional content organized as follows:

- Appendix A presents more related work and discussions:
  - More related work on autoregressive image generation.
  - The reasons for choosing and retaining the LLM architecture.
  - Discussion on other LLM-based visual generation architectures.
  - Discussion on AR-DF's relatedness to Diffusion Forcing and FramePack.
- Appendix B presents a detailed version introducing vanilla 3D RoPE.
- Appendix C presents a detailed version of the AR-DF inference algorithm.
- Appendix D presents more architectural and implementation details.
  - Model architectural details and more training details.
  - Detailed introduction of the token sequence formatting.
  - GPU memory friendly implementation.
- Appendix E presents more analysis and ablation studies.
  - More ablation studies, including
    * The effect of the number of meta MM-RoPEs in MM-RoPE.
    * The effect of the scaling factors in MM-RoPE.
  - More analysis, including
    * `Lumos-1`'s robustness to aspect ratios.
    * `Lumos-1`'s comparison with other RoPE designs.
    * `Lumos-1`'s training resource comparison.
  - More visualizations, including
    * Qualitative visual comparison on text-to-image generation.
    * More text-to-image visualizations.
    * More image-to-video and text-to-video visualizations.
- Appendix F presents more discussions.
  - A road map to unified models for understanding and generation.
  - Limitations and future work.
  - Potential societal impact and safeguards.
- Appendix G presents a disclosure of LLM usage in this paper.

## A  MORE RELATED WORK AND DISCUSSIONS

As a supplement to the main paper, we provide **1)** more related work on autoregressive image generation, **2)** the reasons for choosing LLMs, **3)** the comparison with LLM-based visual generation models and **4)** AR-DF's relatedness to Diffusion Forcing and FramePack.

**Autoregressive image generation.** Spurred by the progress of LLMs, visual generation research is rapidly shifting from diffusion paradigms Rombach et al. (2022); Ho et al. (2020); Zhang et al. (2022); Esser et al. (2024) to an autoregressive one Sun et al. (2024); Tian et al. (2024); Yu et al. (2025); Huang et al. (2025b); Ren et al. (2025); Zhou et al. (2025); Yu et al. (2024); He et al. (2025); Ma et al. (2025); Wang et al. (2025); Ren et al. (2024a), due to its potential of being integrated into unified models Team (2024); Zhuang et al. (2025) with minimal modification. Preliminary research, such as Parti Yu et al. (2022), DALL-E Ramesh et al. (2021), MaskGiT Chang et al. (2022) and LLamaGen Sun et al. (2024), have demonstrated the efficacy of autoregressive generation using discrete tokenizers. MAR Li et al. (2024a) incorporates a diffusion-style objective into autoregressive training, thereby accommodating continuous tokens. VAR Tian et al. (2024) and FAR Yu et al. (2025) reformulate the learning target as next-scale and next-frequency prediction, respectively, improving generation fidelity. Unlike them, we dive into the design of the RoPE technique in visual generation.

**The reasons for choosing and retaining the LLM architecture.** One significant group of unified models, like Chameleon Team (2024) and EMU3 Wang et al. (2024c), adopts the original LLM

architecture to unify understanding and generation tasks. Some hybrid architectures combining an autoregressive transformer and a diffusion head, like BLIP3o-Next Chen et al. (2025), also rely on the autoregressive visual generation capabilities. Although the autoregressive LLM transformers are very capable language models since they are designed for language tasks, the visual generation results are not satisfactory due to the lack of research exploration. We align our model architecture with the LLM so that the insights obtained and techniques designed (e.g., MM-RoPE and AR-DF) in this paper could transfer smoothly to the unified models to come (but we also want to mention that techniques like MM-RoPE can be transferred to other types of visual generation models). This architectural alignment also means that `Lumos-1` is not just a specialized generator; the addition of understanding-oriented data to its pre-training can unlock both generation and comprehension capabilities, which is a promising future work.

**Discussion on other LLM-based unified visual generation architectures.** We briefly compare our model with representative LLM-based unified generative models, including DALL·E Ramesh et al. (2021), LlamaGen Sun et al. (2024), Loong Wang et al. (2024d) and Lumina-mGPT Liu et al. (2024b) in Tab. A1 to clarify the key distinctions. This comparison highlights three key advantages of Lumos-1's design: **1)** Truly unified architecture: Unlike hybrid models like LlamaGen which stitch together a T5 encoder and an LLM (for visual generation), Lumos-1 uses a single, end-to-end LLM backbone. This is a simpler and more elegant design. **2)** Advanced positional encoding (MM-RoPE): Lumos-1 is the only model that incorporates a native 3D-aware RoPE (MM-RoPE) that is also compatible with standard text RoPE used in LLMs. This allows for principled spatiotemporal modeling for visual data, a capability absent in models using 1D, 2D, or non-RoPE embeddings. **3)** Efficient prediction paradigm: Crucially, Lumos-1 is the only model in this comparison to leverage fast, parallelizable mask-based prediction. Other models are constrained by the slow, sequential nature of next-token prediction, which becomes prohibitively inefficient for videos. These targeted architectural choices allow Lumos-1 to function as a fast and versatile image-video generator while being consistent with the core LLM structure.

Table A1: **Comparison of different LLM-based visual generation models.** The table highlights their architecture, position embedding techniques, prediction paradigms, generated media types, and performance on the GenEval benchmark.

| Model | Architecture | Position Embedding | Prediction Paradigm | Generated Media | GenEval |
|---|---|---|---|---|---|
| DALL·E | LLM | sin / cos PE | Next-token prediction (Slow) | Image | – |
| LLamaGen | T5 + LLM (2.9B + 0.8B) | Naïve 2D RoPE | Next-token prediction (Slow) | Image | 0.32 |
| Loong | LLM | 1D RoPE | Next-token prediction (Slow) | Image + Video | – |
| Lumina-mGPT | LLM (7B) | 1D RoPE | Next-token prediction (Slow) | Image | 0.56 |
| Lumos-1 | LLM (1.5B) | MM-RoPE (distributed and scaled 3D RoPE) | Mask prediction (Fast) | Image + Video | 0.725 |

**Discussion on AR-DF's relatedness to Diffusion Forcing and FramePack.** For Diffusion Forcing Chen et al. (2024a) and AR-DF, they are related since both of them are methods to overcome the fundamental *drifting problem* in video generation: A small error made early on gets passed down and magnified over time, causing the final video to "drift" far away from the intended or realistic result. The temporal dependency of the video data, to some extent, leads to the drifting problem since later frames' predictions rely on previous frames. We identify that this is exacerbated by frame-wise loss imbalance since the model tends to optimize those easier tasks, encouraging later frames to rely on "copying" previous frames. The temporal tube masks in AR-DF or the per-frame independent noise in Diffusion Forcing Transformer Song et al. (2025) both aim to ameliorate this issue by reducing the reliance of later frames on previous frames. The inference-time masks in `Lumos-1` also mitigate error accumulation by dropping partially generated context. Our temporal tube masking is particularly suited for video data, as it directly targets the spatial information redundancy between frames. This can be revealed by the experiments in Tab. 5, where we compare our model with a variant trained by following Diffusion Forcing Transformer. We observe that they have similar performance on image generation, while `Lumos-1` performs better on video generation. For FramePack Zhang & Agrawala (2025), it compresses the history in a context with a fixed length. Due to its strategic method for history compression, it focuses more on overcoming the *forgetting problem* (*i.e.*, later frames forget history context), which is not the focus of `Lumos-1`. It also proposes anti-drifting sampling to get rid of drifting. However, it relies on access to future frames.

## B    DETAILED PRELIMINARY OF 3D RoPE

One de facto design of contemporary LLM is RoPE Su et al. (2024), whose overall aim is to encode the absolute position with a rotation matrix while incorporating the explicit relative position dependency in the attention mechanism. This can be formulated as:

$$\langle f_q(\boldsymbol{x}_m, m), f_k(\boldsymbol{x}_n, n) \rangle = g(\boldsymbol{x}_m, \boldsymbol{x}_n, m - n) \tag{8}$$

where $f_q(\boldsymbol{x}_m, m)$ encodes the position $m$ for the embedding $\boldsymbol{x}_m$ to obtain the query feature ($f_k(\boldsymbol{x}_n, n)$ is analogously defined); $g(\boldsymbol{x}_m, \boldsymbol{x}_n, m - n)$ is a function that defines the inner product between the query and key vectors that explicitly encodes the relative position $m - n$. The resultant form of function $f_{\{q,k\}}$ can be formulated as:

$$f_{\{q,k\}}(\boldsymbol{x}_m, m) = \boldsymbol{R}_{\Theta,m}^d \boldsymbol{W}_{q,k} \boldsymbol{x}_m \tag{9}$$

where $\boldsymbol{W}_{q,k}$ is the projection matrix; $\boldsymbol{R}_{\Theta,m}^d$ is the rotary matrix with pre-defined parameters $\Theta = \{\theta_i = \beta^{-2(i-1)/d}, i = [1, 2, ..., d/2]\}$ with $d$ acting as the dimension of the embeddings and $\beta$ acting as the base frequency. In such a formulation, the attention calculation can be rewritten as:

$$f_q(\boldsymbol{x}_m, m)^{\mathrm{T}} f_k(\boldsymbol{x}_n, n) = \boldsymbol{x}_m^{\mathrm{T}} \boldsymbol{W}_q^{\mathrm{T}} \boldsymbol{R}_{\Theta,\tau}^d \boldsymbol{W}_k \boldsymbol{x}_n, \quad \tau = n - m \tag{10}$$

where the detailed formulation of $\boldsymbol{R}_{\Theta,\tau}^d$ can be formulated using a base rotary matrix $R_{\theta,\tau}$, with $\theta$ as the frequency and $\tau$ as the relative position:

$$\boldsymbol{R}_{\Theta,\tau}^d = \begin{bmatrix} R_{\theta_1,\tau} & 0 & \cdots & 0 \\ 0 & R_{\theta_2,\tau} & \cdots & 0 \\ \vdots & \vdots & \ddots & 0 \\ 0 & 0 & \cdots & R_{\theta_{d/2},\tau} \end{bmatrix}, \quad R_{\theta,\tau} = \begin{bmatrix} \cos \tau\theta & -\sin \tau\theta \\ \sin \tau\theta & \cos \tau\theta \end{bmatrix} \tag{11}$$

However, the application of the original RoPE to modeling visual data remains suboptimal considering the spatiotemporal correlation of visual tokens. One popular type of generative models, diffusion models Ho et al. (2022; 2020), improved upon this technique by proposing 3D RoPE that jointly injects spatiotemporal latent coordinates during attention calculation and demonstrates its effectiveness Hong et al. (2022); Kong et al. (2024); Team (2025). If we slightly abuse the annotation by denoting $\boldsymbol{x}_m^{\mathrm{T}} \boldsymbol{W}_q^{\mathrm{T}}$ and $\boldsymbol{W}_k \boldsymbol{x}_n$ as $\boldsymbol{X}_m^{\mathrm{T}}$ and $\boldsymbol{X}_n$, we can write the attention calculation in Eq. (8) based on 3D RoPE as:

$$\boldsymbol{X}_{m,t_s:t_e}^{\mathrm{T}} \begin{bmatrix} R_{\theta_{t_s+1},\tau_t} & \cdots & 0 \\ \vdots & \ddots & \vdots \\ 0 & \cdots & R_{\theta_{t_e},\tau_t} \end{bmatrix} \boldsymbol{X}_{n,t_s:t_e} + \boldsymbol{X}_{m,h_s:h_e}^{\mathrm{T}} \begin{bmatrix} R_{\theta_{h_s+1},\tau_h} & \cdots & 0 \\ \vdots & \ddots & \vdots \\ 0 & \cdots & R_{\theta_{h_e},\tau_h} \end{bmatrix} \boldsymbol{X}_{n,h_s:h_e} + \boldsymbol{X}_{m,w_s:w_e}^{\mathrm{T}} \begin{bmatrix} R_{\theta_{w_s+1},\tau_w} & \cdots & 0 \\ \vdots & \ddots & \vdots \\ 0 & \cdots & R_{\theta_{w_e},\tau_w} \end{bmatrix} \boldsymbol{X}_{n,w_s:w_e} \tag{12}$$

where $\{t_s, t_e\} = \{0, \frac{2}{16}d\}$, $\{h_s, h_e\} = \{\frac{2}{16}d, \frac{5}{16}d\}$ and $\{w_s, w_e\} = \{\frac{5}{16}d, \frac{1}{2}d\}$ denote the start and end dimension index for encoding temporal, height and width relative position; $\boldsymbol{X}_{m,t_s:t_e}^{\mathrm{T}}$ denotes the submatrix extracted from $\boldsymbol{X}_m^{\mathrm{T}}$ using row indices $[2t_s, 2t_e]$; other matrices are similarly defined.

## C    DETAILED AR-DF INFERENCE ALGORITHM

In this subsection, we present the detailed version of the AR-DF inference algorithm. After the text prompt $\boldsymbol{X}_p$ is encoded once and its KV pairs are cached (Lines 1–3), we iterate over the $T$ latent frames (Line 6). Each frame is generated by running $N_{steps}$ iterations (Lines 9–22). During the frame generation process, the model predicts a token distribution, samples tokens only at positions that are still masked, gathers their confidences, and sets the scores of already-revealed tokens to $+\infty$. After adding Gumbel noise, we re-mask the lowest-confidence tokens. As the generation proceeds, fewer tokens are re-masked as the running steps grow and the frame becomes sharper and clearer. The partially revealed frame is then added to the cache before the next frame is generated. Once all frames are finished, they are decoded to RGB space to form the final video.

## D    MORE ARCHITECTURAL AND IMPLEMENTATION DETAILS

**Model architectural details and more training details.** For all of our models, we adopt the Adam Loshchilov & Hutter (2018) optimizer to optimize the model, set the weight decay to 0.1 and

---

**Algorithm 3** AR-DF Inference Procedure

---

**Require:** text prompt $\boldsymbol{X}_p$, trained model $G_\phi$, inference mask ratio $\rho_{inf}$, number of latent frames $T$, number of generation steps $N_{steps}$, number of tokens in a latent frame $N_f$, KV cache $\mathcal{C} = \varnothing$, generated latent list $\mathcal{V} = \varnothing$

1: **Initialize text cache**:
2:      Generate text causal mask $AttnMask^{(p)}$ for $\boldsymbol{X}_p$
3:      $\mathcal{C} \leftarrow G_\phi(\boldsymbol{X}_p, AttnMask^{(p)})$           ▷ Store cache for the prompt
4: **Sample cache mask**:
5:      $\boldsymbol{M}_{inf} \sim \text{Bernoulli}(1 - \rho_{inf})$
6: **for** $t = 1 \rightarrow T$ **do**           ▷ Reused for all frames
7:      Initialize all tokens in $\boldsymbol{X}_v^{(t)}$ as [MASK]
8:      Generate temporal causal mask $AttnMask^{(t)}$ for $\{\boldsymbol{X}_p, \boldsymbol{X}_v^{(t)}\}$
9:      **for** $n = 1 \rightarrow N_{steps}$ **do**
10:          $\boldsymbol{P} \leftarrow G_\phi(\boldsymbol{X}_v^{(t)}, AttnMask^{(t)}, \mathcal{C})$      ▷ Predicted token distribution for all tokens
11:          $sampled\_ids \leftarrow \text{multinomial}(\boldsymbol{P}, 1)$
12:          $sampled\_ids \leftarrow \text{where}(\boldsymbol{X}_v^{(t)} = [\text{MASK}], sampled\_ids, \boldsymbol{X}_v^{(t)})$
13:          $\boldsymbol{Q} \leftarrow \text{Gather}(\boldsymbol{P}, sampled\_ids)$      ▷ Select confidence score of every chosen token
14:          $\boldsymbol{Q} \leftarrow \text{SetKnownInfinity}(\boldsymbol{Q}, \boldsymbol{X}_v^{(t)} \neq [\text{MASK}])$    ▷ Set $+\infty$ where the token was *not* masked
15:          **Mask the lowest confident ($\boldsymbol{Q}$) tokens in *sampled_ids* with randomness:**
16:             Perturb $\boldsymbol{Q}$ with gumbel noise
17:             $\alpha \leftarrow \cos(\frac{\pi}{2} \frac{n}{N_{steps}})$      ▷ Cosine schedule factor in [0,1]
18:             $U \leftarrow \text{floor}(\alpha \times N_f)$      ▷ Number of tokens to mask in this iteration
19:             $low\_conf\_pos \leftarrow \text{SelectLowConfidence}(\boldsymbol{Q}, U)$
20:             $sampled\_ids[low\_conf\_pos] \leftarrow [\text{MASK}]$
21:          $\boldsymbol{X}_v^{(t)} \leftarrow sampled\_ids$
22:      **end for**
23:      Append $\boldsymbol{X}_v^{(t)}$ to $\mathcal{V}$
24:      **Cache partial observation of the generated frame**:
25:      $\widetilde{\boldsymbol{X}}_v^{(t)} = \boldsymbol{M}_{inf} \odot \boldsymbol{X}_v^{(t)} + (1 - \boldsymbol{M}_{inf}) \odot [\text{MASK}]$
26:      $\mathcal{C} \leftarrow G_\phi(\widetilde{\boldsymbol{X}}_v^{(t)}, AttnMask^{(t)}, \mathcal{C})$      ▷ Store cache for the $t$-th frame
27: **end for**
28: Decode $T$ latents $\mathcal{V}$ to RGB space
29: **return** the generated video frames

---

Figure C1: **Detailed inference algorithm with AR-DF.**

set $\beta_1$ and $\beta_2$ to 0.9 and 0.95. Tab. D1 presents the architectural details of three versions of `Lumos-1` and their corresponding training details, including their training batch sizes and their learning rates. For 256p joint training, we use all 60M images and 10M videos. For 384p joint training, we curate a higher-quality subset from 256p training data, consisting of 8M images and 0.6M videos. Following the alternating strategy of MovieGen Polyak et al. (2024), we interleave image and video batches during joint training.

**Detailed token sequence formatting.** The visual tokens and text tokens are interleaved within a sequence Liu et al. (2024b), with the text tokens specifying metadata, including the text prompt, video resolution, video fps and the number of frames in this video. With this design, we train the model using images and videos of varying aspect ratios without resizing them. The text tokens are formatted as "Generate a video with a resolution of $< video\_resolution >$, consisting of $< video\_\#frames >$ frames at $< video\_fps >$ frames per second, according to the following prompt:\n $< text\_prompt >$". The visual tokens are extracted and organized according to the following structured format: "$< video\_start\_token >, < video\_duration\_token >, < video\_fps\_token >, < frame\_tokens >,...,< video\_end\_token >$". $< frame\_tokens >$ contains tokens for each latent frame, which are formatted as: "$< image\_start\_token >, < h\_grid\_token >, < w\_grid\_token >, < image\_content\_tokens >, ..., < image\_end\_token >$", where $< image\_content\_tokens >$ are raster-scan visual tokens with a $< new\_line\_token >$ inserted after each row of visual tokens. To harmonize text and visual tokens, we utilize MM-RoPE, which encodes spatiotemporal coordinates for visual tokens and global positions for text.

**GPU memory friendly implementation.** By default, we leverage flash attention Dao (2023) to accelerate attention calculation and reduce memory overhead during training and inference of

Table D1: **Model architectural and training details.** Batch sizes (*/*) are for images and videos during joint training. Learning rates (*/*) are for 256p and 384p. The 0.5B version is only for fast ablation studies.

| Model | #Params | #Layers | Hidden size | #Heads | Head dim | Batch (256p) | Batch (384p) | Learning rate |
|---|---|---|---|---|---|---|---|---|
| Lumos-1 0.5B | 0.5B | 16 | 1024 | 16 | 64 | 896 / 128 | 192 / 32 | 5e-5 / 2.5e-5 |
| Lumos-1 1B | 1.5B | 16 | 2048 | 32 | 64 | 640 / 96 | 192 / 32 | 5e-5 / 2.5e-5 |
| Lumos-1 3B | 3.6B | 28 | 3072 | 24 | 128 | 768 / 96 | 288 / 48 | 1e-4 / 5.0e-5 |

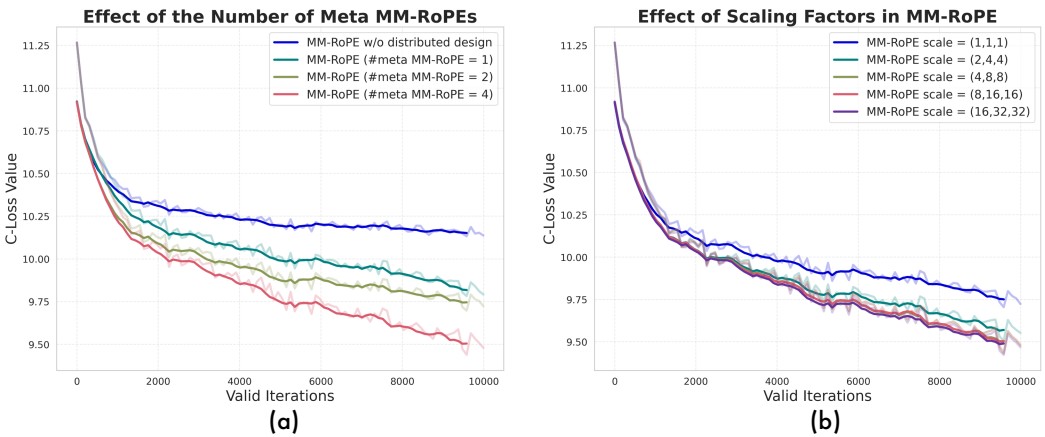

Figure E1: (a) **Effect of the number of meta MM-RoPEs** in MM-RoPE; (b) **Effect of the scaling factors** in MM-RoPE. Both curves are plotted using the 0.5B model trained on videos.

Lumos-1. Moreover, we observe significant amount of GPU memory consumption during the training of a model with a large codebook size. To address it, we eliminate the use of language-related loss (*i.e.*, next-token prediction on texts), reducing the final logit matrix size to match only visual tokens. While text token embeddings, which map the text token index to token embeddings, remain trainable, this focuses the model on video generation. This loss can be added if we target a unified model that is capable of learning within the language modality. Finally, we observe substantially high GPU memory consumption during loss computation over $129k$ token types, which easily leads to the out-of-memory issue. To solve this, we utilize chunked cross-entropy loss, which maintains full softmax accuracy by upcasting and calculating softmax logits one chunk of token sequences at a time. This approach significantly reduces peak memory usage. By default, we set the chunk size to 2,000.

# E MORE ANALYSIS AND ABLATION STUDIES

## E.1 MORE ABLATION STUDIES

**Effect of the number of meta MM-RoPEs in MM-RoPE.** MM-RoPE slices the embedding channels into several meta groups. More groups mean that one kind of information (temporal, height or width) receives a wider and more comprehensive frequency spectrum for modeling. Fig. E1(a) plots the validation loss for a 0.5B model under four settings: **1)** *w/o distributed design*: This setting uses the previous design, which allocates the first $\frac{2}{8}$ channels for temporal modeling and $\frac{3}{8}$ channels for height and width modeling, respectively. **2)** *#meta MM-RoPE=1*: This setting equips a meta MM-RoPE of 64 channels, while maintaining the ratios $(2 : 3 : 3)$ for modeling temporal, height and width information. This variant improves upon the previous one by enabling interleaved height and width channels, which increases the frequency spectrum range for the two spatial dimensions. **3)** *#meta MM-RoPE=2*: This setting equips a meta MM-RoPE of 32 channels. This variant improves the frequency spectrum range for temporal, height and width information compared with the previous one. **4)** *#meta MM-RoPE=4*: This setting is the default design that keeps the number of channels for every meta MM-RoPE minimal (16 channels). This means that the frequency spectrum range is most comprehensive for the temporal, height, or width dimension. These results confirm that widening the frequency spectrum for every dimension through increasing the number of meta MM-RoPEs substantially improves spatiotemporal modeling and overall training effectiveness.

**Effect of the scaling factors in MM-RoPE.** Fig. E1(b) presents the validation loss curves of varying the scaling factors for modeling temporal, height and width positional information in MM-RoPE. Two clear trends emerge: Moving

Table E1: **GenEval score across different aspect ratios** using 1B model.

|  | $448 \times 256$ (7 : 4) | $352 \times 352$ (1 : 1) | $256 \times 448$ (4 : 7) |
|---|---|---|---|
| GenEval | 0.605 | 0.601 | 0.569 |

from $(1, 1, 1) \rightarrow (2, 4, 4) \rightarrow (4, 8, 8)$ steadily lowers the curve, but further enlarging to $(8, 16, 16)$ or $(16, 32, 32)$ yields no additional gain since the three curves almost overlap throughout training. Therefore, a moderate scale of $(4, 8, 8)$ is sufficient to balance vision-language ranges and fully realize the benefit of higher-resolution RoPE, while avoiding unnecessary frequency inflation. We therefore adopt $(4, 8, 8)$ as the default scaling for MM-RoPE.

## E.2 MORE ANALYSIS

**Robustness to aspect ratios.** Although the aspect ratio of training data is mostly 7 : 4, Tab. E1 indicates that `Lumos-1` 1B adapts well to visual generation with different aspect ratios due to the unified codebook design.

**Comparison with other RoPE designs.** To demonstrate the superiority of MM-RoPE, we compare it with five other RoPEs, *i.e.*, M-RoPE Wang et al. (2024b), VideoRoPE Wei et al. (2025), U-RoPE Tang et al., IL-RoPE Liao et al. (2025) and HoPE Li et al. (2025a). We show their design comparison in Tab. E2. We mainly compare these methods on four

Table E2: **Design comparison with other types of RoPE**.

| RoPE Type | Compatiable with Text RoPE | 3D Structure | Comprehensive Frequency Allocation | Strategic Scaling |
|---|---|---|---|---|
| M-RoPE | ✔ | ✔ | ✗ | ✗ |
| U-RoPE | ✔ | ✔ | ✗ | ✗ |
| IL-RoPE | ✗ | ✔ | ✗ | ✗ |
| VideoRoPE | ✔ | ✔ | ✗ | ✔ |
| HoPE | ✔ | ✔ | ✗ | ✔ |
| MM-RoPE | ✔ | ✔ | ✔ | ✔ |

aspects: **1)** their compatibility with text RoPE, so that models can align with contemporary LLMs; **2)** the inclusion of 3D information into RoPE to enable enhanced spatiotemporal modeling; **3)** the comprehensive frequency allocation that enables more balanced local-global context modeling; **4)** the strategic scaling of spatiotemporal indices (or their subsets) to account for different granularities of vision and text information. Based on the above analysis, MM-RoPE holistically incorporates the four properties.

Moreover, we conduct a quantitative comparison in Tab. E3, using our 1B model to perform joint training for 10k steps. Note that M-RoPE is our baseline, equal to eliminating the distributed design and the scaling design from MM-RoPE. U-RoPE improves upon M-RoPE by reallocating high-frequency components for modeling local spatial details, which is useful. IL-RoPE allocating partial channels for modeling spatial information with an interleaved design, while it only allocates partial channels for modeling texts and visual data, hurting the performance. The concurrent work, VideoRoPE, proposes several tech-

Table E3: **Performance comparison with other types of RoPE**. Results are measured on GenEval, Vbench-Overall Consistency (OC), and Vbench-Imaging Quality (IQ).

| RoPE Type | GenEval | Vbench-OC | Vbench-IQ |
|---|---|---|---|
| M-RoPE | 0.310 | 0.122 | 0.350 |
| U-RoPE | 0.402 | 0.165 | 0.423 |
| IL-RoPE | 0.541 | 0.225 | 0.513 |
| VideoRoPE | 0.569 | 0.243 | 0.540 |
| HoPE | 0.570 | 0.246 | 0.545 |
| MM-RoPE | 0.591 | 0.249 | 0.559 |

niques for understanding tasks: low-frequency temporal allocation, diagonal layout and adjustable temporal spacing. We use VideoRoPE's default hyperparameters. We observe that VideoRoPE slightly trails MM-RoPE in our task. We attribute this to the difference in frequency allocation and adjustable temporal spacing. Low-frequency temporal allocation enables more local modeling for spatial features and global modeling for temporal features, while it is important to maintain both local and global modeling for spatiotemporal features, which MM-RoPE performs better at. Moreover, adjustable temporal spacing increases the resolution for better modeling, and we believe that adjusting the hyperparameter in adjustable temporal spacing can increase its performance, which is a future work for VideoRoPE. HoPE improves upon VideoRoPE with a zero-frequency strategy for temporal modeling, therefore slightly outperforming VideoRoPE. It trails MM-RoPE because the frequency spectra that HoPE use for modeling spatiotemporal dependency are not as comprehensive as MM-RoPE.

**Comparison of training resources.** We compiled a detailed comparison of training costs against popular text-to-image and text-to-video models. To ensure a fair comparison, we normalized all

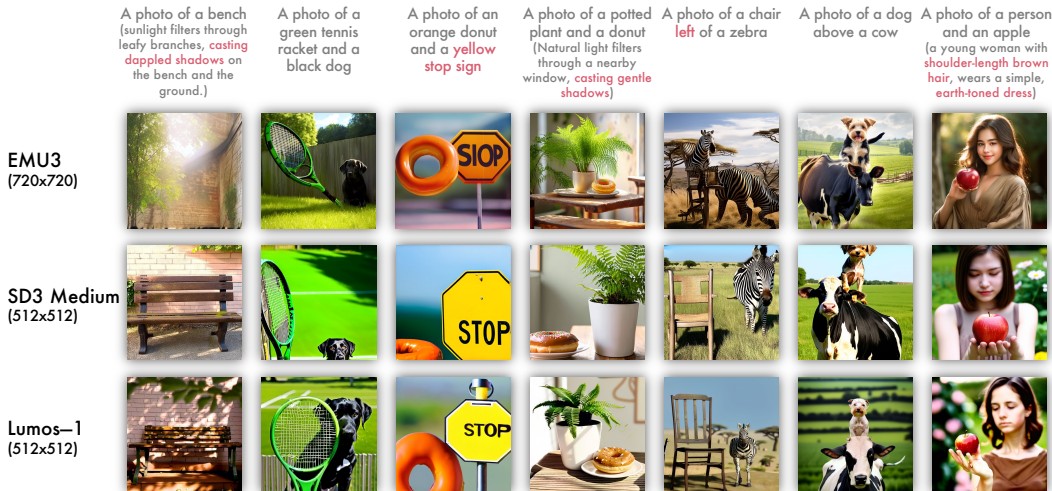

Figure E2: **Visual comparison** of `Lumos-1` with EMU3 and Stabel Diffusion 3 (Medium) on the text-to-image task. We place the simple version of the prompts above the images for reference. Fragments of the detailed prompt are placed in parentheses, with the critical attributes highlighted in red. All images are generated using the detailed prompts for all models.

training times to 8xA100 GPU-Days based on their respective FP16 Tensor Core performance. The results are placed in Tab. E4. We can observe from the table that, compared with text-to-image diffusion models or larger-scale AR models, Lumos-1 is more compute-efficient in terms of training and obtains better performance on GenEval. Compared with Open-Sora Plan v1.2, Lumos-1 achieves similar performance, while reducing the training cost significantly. Unlike diffusion models, which rely on a powerful, external text encoder, Lumos-1 learns language understanding from scratch. Its high efficiency, even with this added learning burden, highlights the effectiveness of our discrete tokenization and MM-RoPE alignment strategy.

### E.3 MORE VISUALIZATIONS

**Qualitative visual comparison on text-to-image generation.** We compare `Lumos-1` with popular text-to-image generation methods in Fig. E2. All models are configured to their default inference settings and use the same version of detailed prompts to ensure quality generation. We can observe that: **1)** Although EMU3 is significantly larger than `Lumos-1`, `Lumos-1` generates images that reproduces fine prompt details more faithfully, such as the dappled shadows on the bench (example 1) and the correct "STOP" text and the yellow color on the road sign (example 3). EMU-3 sometimes drops or warps these details and exhibits occasional anatomical distortions, such as the distorted zebra (example 5) and malformed fingers on the woman holding an apple (example 7). **2)** Stable Diffusion 3 offers slightly crisper textures, which is expected from its continuous tokenizer, but it frequently ignores descriptive constraints, such as the dappled shadows (example 1) and gentle shadows (example 4), and the incorrect dress color (example 7). By contrast, `Lumos-1` meets these constraints while maintaining a convincing overall composition. These examples illustrate that `Lumos-1` delivers strong vision–language alignment compared with both diffusion and autoregressive baselines, while remaining competitive in visual quality.

Table E4: **Comparison of training efficiency** for text-to-image and text-to-video models.

| Method | Params | Time (8×A100 GPU-Days) | GenEval |
|---|---|---|---|
| SD-1.5 | 0.9B + 0.1B | 781.2 | 0.43 |
| SD-2.1 | 0.9B + 0.3B | 1041.6 | 0.50 |
| Dall-E 2 | 4.2B + 1.0B | 5208.3 | 0.52 |
| PixArt-$\alpha$ | 0.6B + 4.7B | 94.1 | 0.48 |
| Lumina-mGPT | 7B | 4488.8 | 0.56 |
| Lumos-1 | 1.5B | **25.9** | **0.725** |

| Method | Params | Time (8×A100 GPU-Days) | VBench |
|---|---|---|---|
| Open-Sora Plan v1.2 | 2.7B + 13B | > 592 | 75.98 |
| Lumos-1 | 1.5B | **85.4** | **78.17** |

**More text-to-image visualizations.** Fig. E3 presents more results generated by `Lumos-1` using descriptive captions. Due to the unified codebook, we can train `Lumos-1` using the original

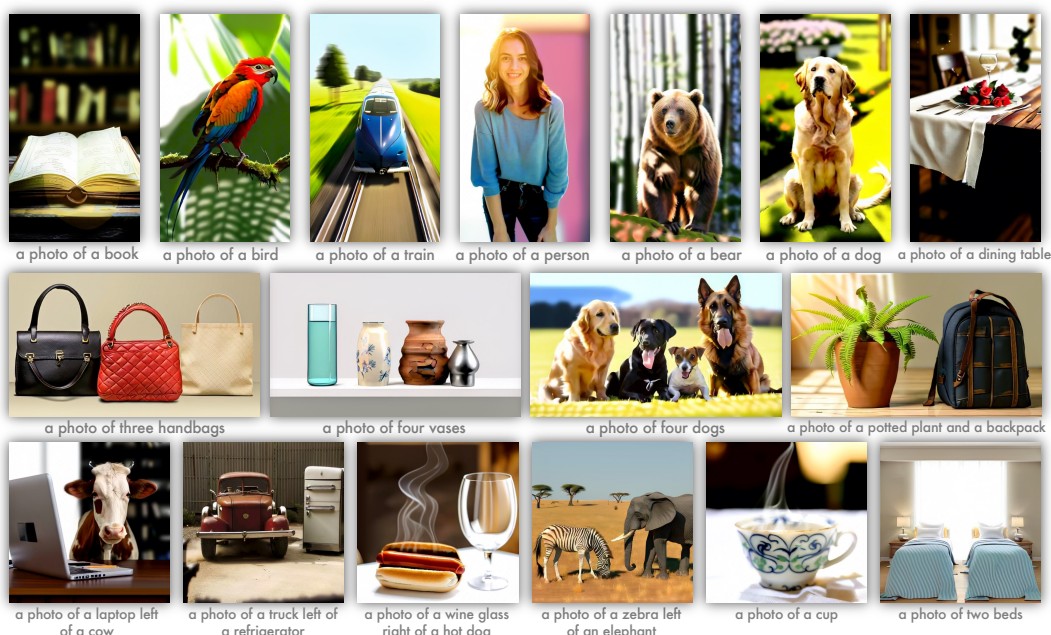

Figure E3: **More text-to-image results** generated by `Lumos-1`. All images are generated using *detailed prompts*, but we only place short prompts here due to space limits. The resolutions for three rows of images are $384 \times 672$, $672 \times 384$ and $512 \times 512$.

resolution of the visual data, enabling generation with different aspect ratios such as $4 : 7$, $7 : 4$, and $1 : 1$, although we do not have a significant amount of data with aspect ratio $4 : 7$. Even though `Lumos-1` is trained from scratch, it transforms lengthy descriptions into complex scenes that are aligned faithfully with the prompts. This can be exemplified by the second row of images, where these images clearly reflect the prompts:

- "*a photo of three handbags*" contains the description: *To the left, a classic black leather handbag with a structured design and gold-tone hardware catches the eye. In the center, a vibrant red quilted handbag with a chain strap adds a pop of color and a touch of luxury. On the right, a casual beige canvas tote bag with simple, clean lines offers a more relaxed aesthetic.*

- "*a photo of four vases*" contains the description: *The first vase on the left is a tall, cylindrical glass vase with a slight teal tint, showcasing clear transparency and a modern design. Next to it is a vintage ceramic vase with an off-white base adorned with hand-painted blue floral patterns, adding a touch of traditional elegance. The third vase is a short, squat terracotta pot with a rustic, earthy texture and visible firing marks, evoking a natural, bohemian feel. Finally, a sleek, metallic silver vase with a geometric shape completes the set, providing a contemporary contrast.*

- "*a photo of four dogs*" contains the description: *To the left is a golden retriever, its fur glowing warmly in the sunlight, with a gentle, friendly gaze. Next to it is a sleek black Labrador, its coat shining with health, head tilted slightly as if listening intently. In the center, a small, energetic Jack Russell terrier bounces with excitement, its white and brown fur clearly defined. On the right, a large German Shepherd sits calmly, its regal posture and dark, expressive eyes adding a dignified presence to the group.*

These qualitative results underscore the strong vision–language alignment achieved by `Lumos-1`.

**More image-to-video and text-to-video visualizations.** Fig. E4 presents more image-to-video results and Fig. E5 presents more text-to-video results generated by `Lumos-1`. The prompts span a wide semantic and dynamical range, containing **1)** close-up animal/human scenes (example 4, 5 in Fig. E5), **2)** fast human and vehicle motion (example 1, 4 in Fig. E4 and example 1, 2 in Fig. E5), **3)** fluid dynamics (example 1, 5 in Fig. E4 and example 2 in Fig. E5), and **4)** complex multi-object collective

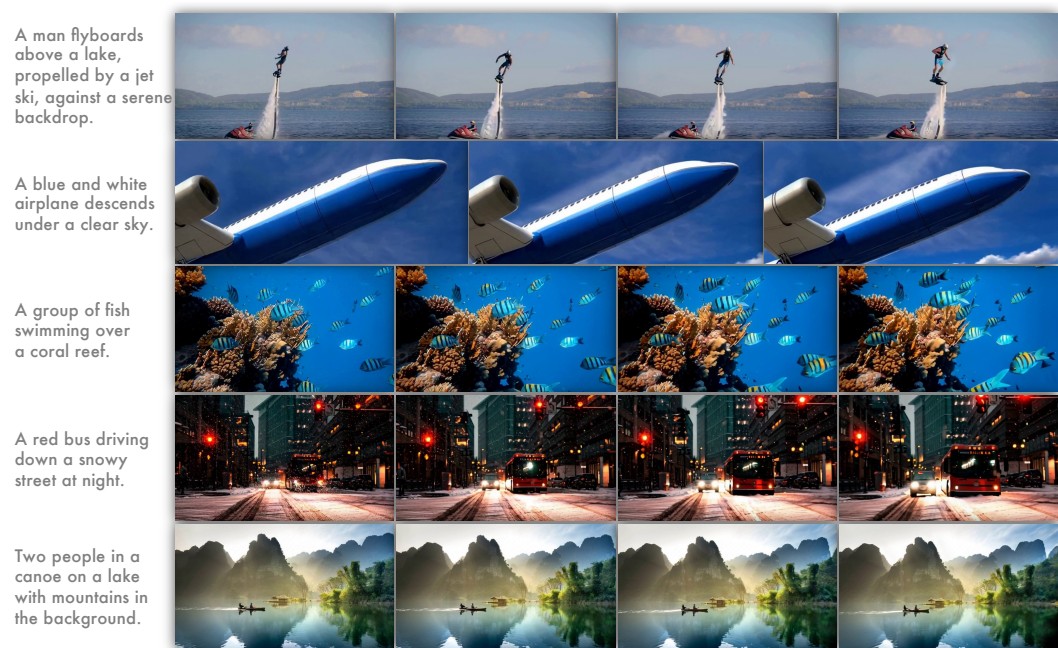

Figure E4: **More image-to-video results** generated by `Lumos-1`. All images are generated using *detailed prompts*, but we only place short prompts here due to space limits. The results encompass videos of resolution $672 \times 384$ and $768 \times 320$.

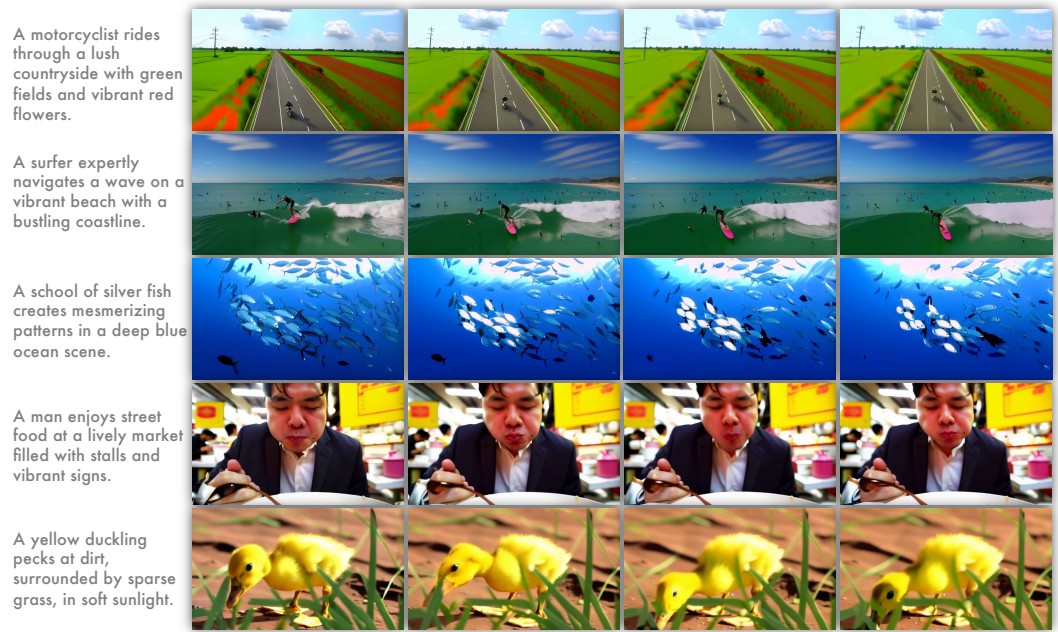

Figure E5: **More text-to-video results** generated by `Lumos-1`. All images are generated using *detailed prompts*, but we only place short prompts here due to space limits. The resolution is $672 \times 384$.

motion (example 3 in Fig. E4 and example 3 in Fig. E5). The results demonstrate that `Lumos-1` can **1)** handle videos of different aspect ratios and **2)** generate single-subject and multi-subject motion that is temporally coherent and physically plausible.

# F    DISCUSSIONS

## F.1    A ROAD MAP TO UNIFIED MODELS FOR UNDERSTANDING AND GENERATION

Building upon our current work, a complete unified model can be achieved through a principled, staged approach:

- **Stage 1: Foundational Model Selection and Codebook Unification.** The starting point is a base model with strong language capabilities. There are two primary paths: **1)** *Path A (Pragmatic Start)*: Initialize with a pre-trained VLM that already possesses strong visual and textual understanding. The main task is then to integrate our generative capabilities by extending its vocabulary with a discrete visual codebook and fine-tuning with our proposed AR-DF and MM-RoPE on generation tasks. **2)** *Path B (Purist, End-to-End)*: Start with a pure LLM. This requires developing a dual-purpose discrete visual tokenizer—one that produces tokens rich enough for both high-fidelity reconstruction (for generation) and high-level semantics (for understanding). This is a challenging but crucial research direction for true end-to-end unification.

- **Stage 2: Multi-Task Data Curation.** A successful unified model requires a diverse, high-quality dataset covering all target modalities and tasks. This involves curating a mixture of: **1)** *Language-only Data*: To maintain and enhance the model's core reasoning and prevent catastrophic forgetting. **2)** *Image/Video-Text Pairs*: For training both understanding (e.g., captioning, VQA) and generation (T2I, T2V) tasks. **3)** *Interleaved Multi-modal Documents*: To teach the model to seamlessly process and generate complex sequences of text and visuals.

- **Stage 3: Joint Multi-Task and Staged Training.** The final step is to train the model to be a versatile multi-task model. This should be done in stages: **1)** *Understanding Foundation (if using Path B)*: First, train the LLM on understanding tasks (e.g., VQA, captioning) to establish a strong vision-language alignment baseline. **2)** *Joint Generation and Understanding Training*: In the main stage, train the model jointly on a mix of understanding and generation objectives. Our proposed AR-DF and MM-RoPE would serve as the core engine for all generative and understanding tasks. By sharing the same transformer backbone, the model is forced to develop a rich, shared representation space where concepts learned from understanding tasks can positively transfer to generation, and vice versa.

## F.2    LIMITATIONS AND FUTURE WORK DISCUSSIONS

We recognize that `Lumos-1`, as an initial endeavor in this area, comes with its limitations. Most prominently, its training corpus (60 million images and 10 million videos) is modest compared with datasets used by recent foundation models Team (2025); Labs (2024), which usually contain billions of samples. Consequently, `Lumos-1` can under-generalize in scenarios that require fine-grained human actions or highly intricate scene dynamics. Considering this, our immediate research plan therefore includes scaling along three axes: **1)** *Data volume and diversity* – expanding both image and video coverage to narrow the generalisation gap. **2)** *Model capacity* – training larger backbones while retaining the MM-RoPE and AR-DF designs. **3)** *Multimodal knowledge infusion* – initializing with strong vision–language models or co-training with visual understanding tasks so the generator can better ground its outputs in world knowledge.

## F.3    POTENTIAL SOCIETAL IMPACTS AND SAFEGUARDS

As the pioneering efforts in autoregressive visual generation, `Lumos-1` can be a significant step towards a large-scale unified model for general visual understanding and generation. Through substantially large-scale training, the obtained model could stimulate intelligence that exceeds existing models trained on unimodal data. The obtained model could serve as a foundation for various downstream applications, thereby reducing the carbon and economic cost that society spends on training various specialized models.

At the same time, unrestricted deployment carries non-trivial risks: the model might be used to fabricate deceptive media, produce disallowed or disturbing visuals, or amplify harmful biases present

in the training data. To mitigate these concerns, we recommend the following safeguards before any real-world release: **1)** *Alignment tuning* - apply preference- or instruction-tuning so that outputs respect human aesthetic and moral preferences Yuan et al. (2024). **2)** *Dual-stage filtering* – i) prompt filters that refuse generation requests involving illicit content, and ii) post-generation classifiers that block or watermark unsafe outputs. **3)** *Red-team evaluation* – continuous adversarial testing to uncover failure modes in new domains or under distribution shift. `Lumos-1` is a research-oriented work, and any broader deployment should proceed only with thorough oversight and evaluation to ensure responsible and ethical use.

# G    DISCLOSURE OF LLM USAGE

We utilized a LLM to assist in the preparation of this manuscript. The primary role of the LLM was to improve the language, clarity, and readability of our writing. All intellectual contributions, including the research ideation, experimental design, data analysis, and the core arguments presented in this paper, are entirely our own. The LLM was used solely as a writing-enhancement tool and did not contribute to the scientific aspects of the work.

