# OpenReview forum: "Lumos-1: On Autoregressive Video Generation with Discrete Diffusion from a Unified Model Perspective"
_ICLR.cc/2026/Conference — ICLR 2026 Poster_

### Official Review · Reviewer_J6ru · 2025-10-19

**Soundness:** 3
**Presentation:** 3
**Contribution:** 3
**Rating:** 6
**Confidence:** 3

**Summary:**

The paper introduces Lumos-1, a unified, LLM-based autoregressive video generator that addresses the central challenge of spatiotemporal positional encoding. To align with the dual nature of video (i.e., spatial bidirectionality and temporal causality), the authors propose two key contributions: 1) MM-RoPE: a distributed, scaled 3D Rotary Position Embedding that preserves standard textual RoPE while interleaving temporal, height, and width channels in repeated "meta" groups to broaden the frequency spectrum. It further scales latent 3D positions by the tokenizer's compression ratio to harmonize visual–text rotary rates, correcting the frequency imbalance inherent in naïve 3D RoPE. AND 2) AR-DF: a mask-based discrete diffusion scheme that replaces next-token decoding under an intra-frame bidirectional and inter-frame causal attention mask. During training, temporal tube masking mitigates late-frame loss collapse caused by spatial redundancy. At inference, consistent partial-context masking combined with KV caching stabilizes quality and motion, outperforming random masking, loss reweighting, and diffusion-forcing baselines.

Lumos-1 is trained on 60M images and 10M videos using 48 GPUs, which achieves competitive text-to-image performance on GenEval compared to diffusion and AR models, and produces image-to-video and text-to-video results on VBench comparable to COSMOS-Video2World and OpenSoraPlan/LTX-Video, despite substantially lower compute and data budgets. The paper also presents extensive, thorough ablation studies.

**Strengths:**

1. The introduction of MM-RoPE is both clear and rigorous. It lays out a coherent, step-by-step narrative: from RoPE fundamentals to naive 3D extensions, diagnosing frequency-spectrum imbalance, and culminating in MM-RoPE, so readers see exactly why each design choice is made. The claims are substantiated by targeted experiments and diagnostics, including validation-loss comparisons across vanilla RoPE, Scheme 1, Scheme 2, and M-RoPE, as well as visual analyses of frequency allocation and rotary speeds that reveal the imbalance.

2. Extensive and convincing ablations: 1) Efficacy of MM-RoPE (showing distribution is primary and scaling complementary); 2) Effectiveness of temporal-tube masks during AR-DF training; 3) Effect of AR-DF inference masks. The "Inference time analysis" suggests substantial speedups for mask-based discrete diffusion with KV caching over next-token decoding.

3. By retaining standard text-side RoPE and an autoregressive transformer, and augmenting them with plug-and-play MM-RoPE for visual tokens and AR-DF for training–inference alignment, the system integrates seamlessly with pretrained LLMs. This composability makes it straightforward to scale to larger multimodal LLMs, leveraging broader pretraining and alignment to improve both understanding and generation.

**Weaknesses:**

1. Despite the title emphasizing "from a unified model perspective", the paper offers relatively little analysis or evidence on the understanding side of a unified system (e.g., text capability, multimodal understanding). While the architecture remains LLM-compatible and preserves text-side RoPE, the work does not demonstrate positive transfer to or from understanding tasks. Given page limits, a full "generation + understanding" suite isn’t necessary, but minimal support would help: Expanding this section with small-scale evaluations or a concrete discussion/roadmap would better align the title with the evidence.

2. Both the fourth and fifth paragraphs of Introduction begin with "To account for the nature of videos," which disrupts the logical flow. I recommend inserting a paragraph before them that explicitly outlines the key considerations for modeling the spatiotemporal visual space. Then, follow with separate paragraphs that elaborate on each aspect, resulting in a clearer and more coherent structure.

3. The current comparisons do not include 2025 state-of-the-art models, which weakens the empirical case. Please add or reference 2025 baselines (both AR and diffusion/hybrid) with matched settings where possible.

**Questions:**

1. Does the design in AR-DF affect long-form video generation, especially complex scenarios like shot transitions, and metrics related to dynamics?

2. I recommend providing playable example videos on the project page to enable more effective qualitative visual comparisons.

---

> ### Author Response · Authors · 2025-11-25
> **Response to Reviewer  J6ru: Part I**
>
> Thank you for your insightful review.
> We're glad you found merit in the novelty and rigor of designing MM-RoPE, the comprehensive experiments and ablation studies to demonstrate the usefulness of all the proposed components, and the potential of integrating Lumos-1 into contemporary unified models.
> We have carefully considered your questions and hope our responses below provide the necessary clarification.
> Please do not hesitate to ask if anything is unclear.
> (W denotes weakness, Q denotes question, and A denotes answer.)
>
> > W1. Expanding this section with small-scale evaluations or a concrete discussion/roadmap to better align the title with the evidence.
>
> **W1A1:**
> Thank you for this constructive feedback.
> Our primary contribution in this work is to demonstrate that high-fidelity video generation can be seamlessly and efficiently integrated into a unified LLM architecture with minimal modifications, rather than to claim that our model is capable of performing general understanding tasks by training only on generation tasks.
> We see this as a critical step on the path towards a fully unified model, following models like Chameleon [1] and EMU-3 [2], which use a unified discrete codebook and process tokens using one LLM.
>
> We would also like to gently highlight that the strong performance on T2I, T2V and I2V benchmarks inherently demonstrates a significant level of language understanding.
> To generate a video/image that accurately reflects a complex prompt, the model must first comprehend the entities, actions, and relationships described.
> Our model learns this alignment from scratch, underscoring the effectiveness of the unified framework.
>
> To fully address your concern and better align our paper with its title, we provide a concrete roadmap for extending our work into a model with both generation and understanding capabilities.
>
> - *Stage 1: Foundational Model Selection and Codebook Unification.*
> The starting point is a base model with strong language capabilities. There are two primary paths:
> *1) Path A*: Initialize with a pre-trained VLM that already possesses strong visual and textual understanding. The main task is then to integrate our generative capabilities by extending its vocabulary with a discrete visual codebook and fine-tuning with our proposed AR-DF and MM-RoPE on generation tasks.
> *2) Path B*: Start with a pure LLM. This requires developing a dual-purpose discrete visual tokenizer—one that produces tokens rich enough for both high-fidelity reconstruction (for generation) and high-level semantics (for understanding). This is a challenging but crucial research direction for true end-to-end unification.
>
> - *Stage 2: Multi-Task Data Curation.*
> A successful unified model requires a diverse, high-quality dataset covering all target modalities and tasks. This involves curating a mixture of: *1)* Language-only Data: To maintain and enhance the model's core reasoning and prevent catastrophic forgetting. *2)* Image/Video-Text Pairs: For training both understanding (e.g., captioning, VQA) and generation (T2I, T2V) tasks. *3)* Interleaved Multi-modal Documents: To teach the model to seamlessly process and generate complex sequences of text and visuals.
>
>
> - *Stage 3: Joint Multi-Task and Staged Training.*
> The final step is to train the model to be a versatile multi-task model. This should be done in stages:
> *1)* Understanding Foundation (if using Path B): First, train the LLM on understanding tasks (e.g., VQA, captioning) to establish a strong vision-language alignment baseline.
> *2)* Joint Generation and Understanding Training: In the main stage, train the model jointly on a mix of understanding and generation objectives. **Our proposed AR-DF and MM-RoPE would serve as the core engine for all generative and understanding tasks.** By sharing the same transformer backbone, the model is forced to develop a rich, shared representation space where concepts learned from understanding tasks can positively transfer to generation, and vice versa.
>
> **Action Taken:**
> To better align our paper with its title and your feedback, we have added this discussion and roadmap to Appendix F of our revised manuscript.

---

> ### Author Response · Authors · 2025-11-25
> **Response to Reviewer J6ru: Part II**
>
> > W2. Add content before fourth and fifth paragraphs to make the flow of ideas smooth.
>
> **W2A2:**
> Thank you for this constructive suggestion.
> We agree that adding content before these two paragraphs could improve the logical flow.
> The third paragraph was originally written for this purpose; therefore, we revised this paragraph, following your recommendation, aiming to improve its clarity and coherence.
> The revised paragraph is marked in Red in the revised manuscript.
> In the revised paragraph, we outline the two fundamental challenges of adapting LLMs for video generation: 1) inadequate RoPE representation and 2) an inefficient and ineffective prediction paradigm.
> The two subsequent paragraphs are now dedicated to explaining how our proposed methods, MM-RoPE and AR-DF, directly address each of these challenges, respectively.
> We believe this new structure provides a much clearer narrative and we appreciate your feedback.
>
>
>
> > W3. Add or reference 2025 baselines (both AR and diffusion/hybrid) with matched settings where possible.
>
> **W3A3:**
> Thank you for this suggestion.
> To address this, we have updated our paper (Tables 1 and 2) to include comprehensive comparisons with several leading 2025 models for both image and video generation.
> For a fair and direct comparison, we prioritized models that, like Lumos-1, utilize discrete representations, which is a fairer comparison.
>
> For video generation, we added VideoMAR (NeurIPS 2025) [3] in Table 2, which uses continuous representations.
> For image generation, which is more popular in the research field, we add SANA-1.5 (ICML 2025) [4], Meissonic (ICLR 2025) [5], Muddit (ICLR 2026 submission) [6], UniDisc (arXiv 2025) [7], D-DiT (CVPR 2025) [8], MMaDA (NeurIPS 2025) [9], Lumina-mGPT 2.0 (arXiv 2025) [10] and Show-o2 (NeurIPS 2025) [11] in Table 1 for comparison.
>
> Most of them use discrete tokenizers like Lumos-1 (except SANA-1.5, VideoMAR and Show-o2).
> We also added a new column denoting the generation representation type (whether the representation for generation is continuous or discrete).
> We can observe from the table that, compared with these added methods, Lumos-1 is still a competitive method, even if our model is trained from scratch, without needing a massive external text encoder.
>
>
> > Q1. Does the design in AR-DF affect long-form video generation, especially complex scenarios like shot transitions, and metrics related to dynamics?
>
> **Q1A1:**
> Thank you for this insightful question. We confirm that the design of AR-DF is compatible with generating long videos, including those with complex dynamics like shot transitions.
>
> The generation process would follow a standard autoregressive, sliding-window approach:
>
> - *Standard Autoregressive Generation*: For videos up to the model's trained context length, generation proceeds exactly as described in Figure 3 of the paper. The model generates one frame at a time, conditioning on the previously generated frames held in a KV cache.
>
> - *Long-Form Generation (Sliding Window)*: To generate videos longer than the trained context, a sliding-window mechanism should be employed: The information corresponding to the oldest frame is dropped from the cache to make room for the newly generated frame's state, once the KV cache is longer than the trained context. This ensures the model always conditions on a fixed-length history.
>
> Regarding your specific concerns:
> *1)* Dynamics and Quality: This sliding-window approach is a useful technique for LLMs and AR models and is known to maintain high-quality generation and local temporal dynamics, as the model is always operating within its trained context length.
> *2)* Complex Scenarios (Shot Transitions): The model's ability to generate complex events like shot transitions is primarily a function of the training data (like diverse examples of cuts, fades, and other transitions), not the AR-DF algorithm itself. But AR-DF's training design with temporal tube masks enhances temporal modeling, which should aid in learning such dynamics.

---

> ### Author Response · Authors · 2025-11-25
> **Response to Reviewer J6ru: Part III**
>
> > Q2. I recommend providing playable example videos on the project page to enable more effective qualitative visual comparisons.
>
> **Q2A2:**
> Many thanks for this useful suggestion.
> We are working on an organized project page and placing our generated videos on it for better visualization.
> This page will be made public and linked in the camera-ready version of the paper upon acceptance.
> We believe this will provide the community with a good resource for evaluating our work.
>
>
> **Reference**:
> [1] Chameleon: Mixed-Modal Early-Fusion Foundation Models.
> [2] Emu3: Next-Token Prediction is All You Need.
> [3] VideoMAR: Autoregressive Video Generation with Continuous Tokens.
> [4] Sana 1.5: Efficient scaling of training-time and inference-time compute in linear diffusion transformer.
> [5] Meissonic: Revitalizing masked generative transformers for efficient high-resolution text-to-image synthesis.
> [6] Muddit: Liberating generation beyond text-to-image with a unified discrete diffusion model.
> [7] Unified multimodal discrete diffusion.
> [8] Dual diffusion for unified image generation and understanding.
> [9] Mmada: Multimodal large diffusion language models.
> [10] Lumina-mGPT 2.0: Stand-Alone AutoRegressive Image Modeling.
> [11] Show-o2: Improved Native Unified Multimodal Models.

---

### Official Review · Reviewer_uiBr · 2025-10-31

**Soundness:** 3
**Presentation:** 4
**Contribution:** 4
**Rating:** 6
**Confidence:** 3

**Summary:**

This paper introduces Lumos-1, an autoregressive video generator based on an LLM architecture with minimal modifications, which aims to create a unified multimodal model. A key achievement is the demonstration of video generation using LLM architecture, which paves the way for a truly unified foundation model and eliminates the need for external text encoders.

Additionally, Lumos-1 incorporates two innovations:
1.  It proposes MM-ROPE to address imbalanced frequency spectrums in 3D RoPE, enhancing spatiotemporal correlation modeling via distributed channel allocation and scaled 3D positions.
2. Lumos-1 employs autoregressive discrete diffusion forcing (AR-DF) to mitigate frame-wise loss imbalance and spatial information redundancy in autoregressive video generation training.

The paper provides detailed descriptions of the methods employed, thorough experimental evaluations against robust benchmarks, and insightful ablation studies that effectively demonstrate the merits of the innovations. However, Lumos-1 is trained from scratch despite a high structural similarity with Llama, so the model needs to learn both language and vision simultaneously, which can introduce instability and inefficiency during training.

In summary, this is an innovative work with significant community impact, and I appreciate its architectural exploration; however, I remain concerned about its training efficiency and stability, and I am curious about its performance with LLM initialization.

**Strengths:**

1. This work proposes Lumios, which demonstrates the effectiveness of video generation using LLM architecture, paving the way for a truly unified foundation model and eliminating the need for external text encoders.
2. It proposes MM-ROPE to address imbalanced frequency spectrums in 3D RoPE, enhancing spatiotemporal correlation modeling via distributed channel allocation and scaled 3D positions.
3. Lumos-1 employs autoregressive discrete diffusion forcing (AR-DF) to mitigate frame-wise loss imbalance and spatial information redundancy in autoregressive video generation training.
4. Comprehensive experiments and complete ablation studies validate the advantage of MM-ROPE and AR-DF, and the effectiveness of LLM-based video generation.

**Weaknesses:**

1. Although structurally similar to Llama, Lumos-1 is trained from scratch, requiring simultaneous learning of language and vision, which may lead to training instability and inefficiency. The validation curve in Figure 7(a) supports concerns about the instability. Furthermore, as a foundation model, the full cost of the training, particularly in terms of time, is not disclosed, which is a cause for concern in terms of the inefficiency of the training.
2. The paper claims minimal structural modification from LLM for a unified text-visual model, but it lacks a comprehensive structural comparison with other established unified generative models or LLM-based autoregressive models, making the claimed advantage less substantiated.

**Questions:**

1. Since the model structure is highly similar to LLM, why not initialize the parameters with pretrained Llama?
2. Based on the experimental results, the unified architecture improves semantic consistency significantly, yet the visual quality still lags behind some baseline methods. Have you attempted to explain this trade-off or conducted a deeper analysis of its causes?

---

> ### Author Response · Authors · 2025-11-25
> **Response to Reviewer uiBr: Part I**
>
> Thank you for your insightful review.
> We're glad you found merit in the significance of Lumos-1 in the era of unified models, the novelty of MM-RoPE, the efficacy of AR-DF, and the comprehensive experiments and ablation studies on three visual generation tasks.
> We have carefully considered your questions and hope our responses below provide the necessary clarification.
> Please do not hesitate to ask if anything is unclear.
> (W denotes weakness, Q denotes question, and A denotes answer.)
>
>
>
> > W1. The possible training instability and inefficiency issue.
>
> **W1A1**:
> Thanks for raising these points. We would like to address each concern separately.
>
> **On Training Stability:**
> Regarding the training instability that you mentioned in Figure 7(a), we think this might be a misunderstanding since Figure 7(a) shows the importance of inference-time masks in AR-DF, not a depiction of the training process.
> To observe the validation loss curve during training, we suggest referring to Figure 6(b).
> As shown, the learning curve is smooth, without the loss fluctuating significantly.
> We attribute this stability to our unified architecture, where both language and vision are represented as discrete tokens.
> This shared modality facilitates a more direct and stable alignment process compared to aligning disparate representations.
> We are happy to provide further details if we misunderstood your question.
>
> **On Training Efficiency:**
> To address your concern about efficiency, we have compiled a detailed comparison of training costs against popular text-to-image and text-to-video models.
> To ensure a fair comparison, we normalized all training times to 8xA100 GPU-Days based on their respective FP16 Tensor Core performance.
>
> | Method | Params | Time (8×A100 GPU-Days) | GenEval |
> |:-|:-:|:-:|:-:|
> | SD-1.5   | 0.9B + 0.1B | 781.2  | 0.43 |
> | SD-2.1   | 0.9B + 0.3B | 1041.6 | 0.50 |
> | Dall-E 2 | 4.2B + 1.0B | 5208.3 | 0.52 |
> | PixArt-α | 0.6B + 4.7B | 94.1   | 0.48 |
> | Lumina-mGPT     | 7B   | 4488.8 | 0.56 |
> | Lumos-1  | 1.5B        | 25.9   | 0.725 |
>
> | Method | Params | Time (8×A100 GPU-Days) | VBench |
> |:-|:-:|:-:|:-:|
> | Open-Sora Plan v1.2  | 2.7B + 13B | > 592   | 75.98 |
> | Lumos-1  | 1.5B      | 85.4   | 78.17 |
>
> We can observe from the table that, compared with text-to-image diffusion models or larger-scale AR models, Lumos-1 is more compute-efficient in terms of training and obtains better performance on GenEval.
> Compared with Open-Sora Plan v1.2, Lumos-1 achieves similar performance, while reducing the training cost significantly.
> Unlike diffusion models which rely on a powerful, external text encoder, Lumos-1 learns language understanding from scratch.
> Its high efficiency, even with this added learning burden, highlights the effectiveness of our discrete tokenization and MM-RoPE alignment strategy.
> We hope this data clarifies that Lumos-1 is both a stable and highly efficient training paradigm.
>
> **Action Taken:** We have added this training efficiency analysis to Appendix E.2.

---

> ### Author Response · Authors · 2025-11-25
> **Response to Reviewer uiBr: Part II**
>
> > W2. Comparison with other established unified generative models or LLM-based autoregressive models.
>
> **W2A2**:
> Thank you for this excellent point.
> To address your concern, we briefly compare our model with representative LLM-based unified generative models, including DALL·E [1], LlamaGen [2], Loong [3] and Lumina-mGPT [4] in the table below to clarify the key distinctions.
> Our core argument is that Lumos-1 achieves superior versatility and efficiency while adhering more closely to a standard LLM architecture than its predecessors.
>
> | Model | Architecture | Position Embedding | Prediction Paradigm | Generated Media | GenEval |
> |:-|:-:|:-:|:-:|:-:|:-:|
> | DALL·E   | LLM      | sin / cos PE  | Next-token prediction (Slow) | Image | - |
> | LLamaGen | T5 + LLM (2.9B+0.8B) | Naïve 2D RoPE | Next-token prediction (Slow) | Image | 0.32 |
> | Loong    | LLM      | 1D RoPE | Next-token prediction (Slow)       | Image + Video | - |
> | Lumina-mGPT | LLM (7B)   | 1D RoPE | Next-token prediction (Slow)       | Image | 0.56 |
> | Lumos-1     | LLM (1.5B)   | MM-RoPE (distributed and scaled 3D RoPE)     | Mask prediction (Fast) | Image + Video | 0.725 |
>
> This comparison highlights three key advantages of Lumos-1's design:
>
> - *Truly Unified Architecture*: Unlike hybrid models like LlamaGen which stitch together a T5 encoder and an LLM (for visual generation), Lumos-1 uses a single, end-to-end LLM backbone. This is a simpler and more elegant design.
> - *Advanced Positional Encoding (MM-RoPE)*: Lumos-1 is the only model that incorporates a native 3D-aware RoPE (MM-RoPE) that is also compatible with standard text RoPE used in LLMs. This allows for principled spatiotemporal modeling for visual data, a capability absent in models using 1D, 2D, or non-RoPE embeddings.
> - *Efficient Prediction Paradigm*: Crucially, Lumos-1 is the only model in this comparison to leverage fast, parallelizable mask-based prediction. Other models are constrained by the slow, sequential nature of next-token prediction, which becomes prohibitively inefficient for videos.
>
> These targeted architectural choices allow Lumos-1 to function as a fast and versatile image-video generator while being consistent with the core LLM structure.
>
> **Action Taken:**
> In the original submission, we have a discussion in Appendix A to compare with LLM-based visual generation methods like LlamaGen and DALL·E.
> In the revised version, we updated that discussion to include more methods mentioned above.
>
> > Q1. The possibility of initializing the parameters with pretrained Llama.
>
> **Q1A1:**
> Many thanks for raising this question.
> While initializing from a pretrained LLM is indeed a promising direction, we trained Lumos-1 from scratch due to practical considerations.
> We utilized the LLM architecture and text codebook from the popular Llama-2 family.
> However, the smallest version of language pre-trained Llama-2 is of 7B size, which is too large for us to train and explore various design choices.
> To maintain the language capability and prevent forgetting textual knowledge, it also requires additional language data co-training, which further expands the scale of the training dataset.
>
> Despite these constraints, our work demonstrates a crucial and encouraging result: a highly effective visual generative model can be trained from scratch with a unified LLM architecture.
> Our strong benchmark results on T2I, I2V, and T2V tasks show that direct vision-language alignment with our proposed methods is highly effective on its own.
> We believe future work using larger sizes and with language co-training during the fine-tuning phase will further boost the performance.

---

> > ### Author Response · Authors · 2025-11-25
> > **Response to Reviewer uiBr: Part III**
> >
> > > Q2. Explanation and analysis of the trade-off between semantic consistency and visual quality.
> >
> > **Q2A2:**
> > Many thanks for raising this insightful question. We will break down the cause of this trade-off.
> >
> > *Reason for High Semantic Consistency:* A unified architecture excels at semantic consistency because all the visual tokens and language tokens are structured to be discrete in a shared codebook.
> > This allows the LLM to perform direct, powerful cross-modal alignment in a unified latent space, naturally leading to strong semantic coherence.
> > Semantic consistency is at the semantic level, which is abstractive, without focusing too much on the fine-grained local details.
> >
> >
> > *The Challenge of Visual Quality:* However, visual quality focuses on fine-grained details, which are heavily determined by the reconstruction capability of the visual tokenizers.
> > Usually, the reconstruction capability of discrete visual tokenizers trails the continuous ones.
> > Consider the COSMOS discrete tokenizer [5] we use in this paper:
> >
> > | Tokenizer Type | PSNR | SSIM| rFVD |
> > |:-|:-:|:-:|:-:|
> > | Discrete (4x8x8) | 28.81 | 0.818 | 37.36 |
> > | Continuous (4x8x8) | 32.80 | 0.900| 15.93 |
> >
> > This data shows that the choice of a discrete tokenizer inherently puts a ceiling on the visual quality at the tokenizer level.
> > However, our paper's core contribution is in designing a highly effective and semantically coherent generative model.
> > Our architecture can benefit from the rapid, ongoing advancements in discrete visual tokenization.
> > As better tokenizers emerge, the visual quality of Lumos-1 will improve without any changes to our model's architecture, while retaining its superior semantic consistency.
> >
> >
> > **Reference**:
> > [1] Zero-shot text-to-image generation.
> > [2] Autoregressive model beats diffusion: Llama for scalable image generation.
> > [3] Loong: Generating minute-level long videos with autoregressive language models.
> > [4] Lumina-mgpt: Illuminate flexible photorealistic text-to-image generation with multimodal generative pretraining.
> > [5] Cosmos World Foundation Model Platform for Physical AI.

---

### Official Review · Reviewer_NXEV · 2025-10-31

**Soundness:** 3
**Presentation:** 3
**Contribution:** 2
**Rating:** 4
**Confidence:** 5

**Summary:**

The paper presents Lumos-1, a unified autoregressive video generation model built upon a LLM architecture, designed to address key limitations of existing autoregressive video generators. To overcome decoding inefficiency, the authors integrate masked diffusion with causal attention. They further introduce MM‑RoPE that preserves standard 1D RoPE for text while effectively capturing spatiotemporal correlations in video. Additionally, the model mitigates error accumulation through a specialized form of diffusion forcing that shares both timestep and mask perturbation across frames. Lumos-1 achieves performance comparable to models such as Emu3, Cosmos1, and OpenSoraPlan v1.3.

**Strengths:**

- Lumos-1 demonstrates that a pure LLM architecture can be employed for autoregressive video generation.
- Lumos-1 addresses the frequency limitations of the original M-RoPE.
- Lumos-1 proposes AR-DF scheme to mitigate the training-inference inconsistency.

**Weaknesses:**

- **Reconsidering the Masking Strategy of AR-DF.** Diffusion forcing[1] models the conditional probabilities between frames by adding independent noise to each frame. However, due to the issue of information leakage in mask-based approaches, AR-DF can only apply the same noise to all frames, which deviates from the core idea of diffusion forcing. In my view, AR-DF is more like a form of data augmentation or an exploration of better masking strategies for video generation. I believe the authors need to reconsider and clarify their masking strategy.
- **Regarding the novelty of MM-RoPE.** I consider that interleaved frequency allocation (IL-RoPE) was first introduced in Mogao[2], the authors should cite this work and provide a comparative analysis between MM-RoPE and IL-RoPE.
- **Suboptimal performance of unified architechture.** Although Lumos-1 employs architecture without a text encoder, its text-to-image and text-to-video performance remains suboptimal.  The authors could substantially improve performance to better align with the promise of its unified architecture. Moreover, some of the reported baselines appear outdated or unfair, for instance, Show-o[3]’s GenEval score is 0.68 without rewritting, and NOVA[4] achieves a VBench score of 80.12 using long prompts.

- [1] Boyuan Chen, Diego Marti Monso, Yilun Du, Max Simchowitz, Russ Tedrake, and Vincent Sitzmann. Diffusion forcing: Next-token prediction meets full-sequence diffusion. arXiv preprint arXiv:2407.01392, 2024a.
- [2] Chao Liao, Liyang Liu, Xun Wang, Zhengxiong Luo, Xinyu Zhang, Wenliang Zhao, Jie Wu, Liang Li, Zhi Tian, and Weilin Huang. Mogao: An omni foundation model for interleaved multi-modal generation. arXiv preprint    arXiv:2505.05472, 2025.
- [3] Jinheng Xie, Weijia Mao, Zechen Bai, David Junhao Zhang, Weihao Wang, Kevin Qinghong Lin, Yuchao Gu, Zhijie Chen, Zhenheng Yang, and Mike Zheng Shou. Show-o: One single transformer to unify multimodal understanding and generation. In ICLR, 2025.
- [4] Haoge Deng, Ting Pan, Haiwen Diao, Zhengxiong Luo, Yufeng Cui, Huchuan Lu, Shiguang Shan, Yonggang Qi, and Xinlong Wang. Autoregressive video generation without vector quantization. In ICLR, 2025.

**Questions:**

- See weaknesses.
- In short, the work presents Lumos-1 model for autoregressive video generation,  the contributions of this are work are not sufficiently substantiated by the experimental results:
1. A unified architecture for visual generation with strong performance.

2. Asynchronous AR-DF for mitigating the training-inference inconsistency.

3. The difference between IL-RoPE and MM-RoPE on interleaved frequency allocation.

If the authors provide stronger model performance or a more thorough analysis of the weaknesses, I would be happy to raise my score.

---

> ### Author Response · Authors · 2025-11-25
> **Response to Reviewer NXEV: Part I**
>
> Thank you for your insightful review.
> We're glad you found merit in the superiority of using LLMs for video generation, the novelty of MM-RoPE, and the efficacy of AR-DF.
> We have carefully considered your questions and hope our responses below provide the necessary clarification.
> Please do not hesitate to ask if anything is unclear.
> (W denotes weakness, Q denotes question, and A denotes answer.)
>
>
> > W1. Reconsidering the masking strategy of AR-DF.
>
> **W1A1:**
> Thank you for this insightful question.
> We agree that the masking strategy in AR-DF is a deliberate departure from the frame-independent noise used in the original diffusion forcing. This change is central to AR-DF.
> We would like to respectfully clarify the distinction between AR-DF and data augmentation.
>
> - Data augmentation typically aims to increase the diversity of the training set by generating new, plausible training samples (e.g., via flips, crops, or adding unstructured noise). Its goal is to improve model generalization by expanding the data distribution and adding more stochasticity.
>
> - AR-DF, in contrast, does not add more stochasticity. Instead, it imposes a specific structure on the training objective itself. By using tube masking, we constrain the noise schedule to be temporally correlated. This reduces the stochasticity compared to frame-independent noise.
>
> This is a principled design choice. We trade the frame-level independence of diffusion forcing for a structure that more accurately reflects the autoregressive nature of video.
> This ensures a more balanced learning in different frames so that we can obtain better videos.
> As shown in Table 4 of the original manuscript (also presented below), our strategy can outperform the original diffusion forcing, especially in video metrics.
>
> | Training Methods | GenEval | Vbench-OC | Vbench-IQ |
> |:-|:-:|:-:|:-:|
> | Diffusion Forcing | 0.590 | 0.241 | 0.540 |
> | AR-DF | 0.591 | 0.249 | 0.559 |
>
> Therefore, AR-DF is better understood not as data augmentation, but as a structured training objective in video AR models (or an exploration, as you mentioned).
>
>
> > W2: A comparison with IL-RoPE.
>
> **W2A2:**
> Thank you for this suggestion.
> Although IL-RoPE partially incorporates the interleaved design, MM-RoPE are still significantly different from it.
> We list some key designs in the table below to highlight the difference.
>
> | RoPE type | Compatiable with text RoPE | 3D structure | Comprehensive frequency allocation | Strategic scaling |
> |:-|:-:|:-:|:-:|:-:|
> | RoPE        |  ✔  |  ✗  |  ✗  |  ✗  |
> | M-RoPE      |  ✔  |  ✔  |  ✗  |  ✗  |
> | IL-RoPE [1] |  ✗  |  ✔  |  ✗  |  ✗  |
> | MM-RoPE     |  ✔  |  ✔  |  ✔  |  ✔  |
>
> *1)* First of all, the original text RoPE would utilize all the frequency spectra for modeling textual dependency, while IL-RoPE only uses half of the spectra (the highest one quarter and the lowest one quarter), limiting the learning capacity and reducing its alignment with the original LLM.
> *2)* Secondly, IL-RoPE's interleaved design is constrained to the first three quarters of the frequency spectra, while MM-RoPE's interleaved design expands the full frequency spectra, achieving comprehensive frequency allocation.
> *3)* Finally, we observe that the latent space index, which is used in IL-RoPE, is suboptimal in performance. We attribute this to inferior spatiotemporal modeling and vision-language alignment. We added scaling factors to the visual indices, which shows improvement.
> This can be validated by the experiment below, which fine-tunes Lumos-1 with image-video data for 10k steps using two RoPEs.
> The performance gap in the table shows MM-RoPE's superiority.
>
> | RoPE Type | GenEval | Vbench-OC | Vbench-IQ |
> |:-|:-:|:-:|:-:|
> | IL-RoPE   | 0.541 | 0.225 | 0.513 |
> | MM-RoPE   | 0.591 | 0.249 | 0.559 |
>
> **Action Taken:** In the revised manuscript, we have added IL-RoPE in the Related Work section and added experiments on IL-RoPE in Table 7 of the main paper (short version) and Appendix E2 (detailed version).

---

> ### Author Response · Authors · 2025-11-25
> **Response to Reviewer NXEV: Part II**
>
> > W3: Suboptimal performance and out-of-date baselines.
>
> **W3A3:**
> Many thanks for raising the point, and we apologize for the outdated baseline results.
> We *updated their results* in their corresponding tables to reveal their up-to-date performance.
>
> Regarding the suboptimal performance, one of the reasons that leads to this issue lies in the inherent advantage of continuous tokenizers over their discrete counterparts, rather than a failure of the unified architecture itself.
>
> To enable a fairer comparison, we add a column to indicate the representation type used for generation, whether it is continuous or discrete, and reorganize the table.
> Moreover, we added more methods (public in 2025) using discrete tokenizers similar to Lumos-1 in Table 1.
> Compared with these methods like EMU3 8B and MMaDA 8B, Lumos-1 is still a competitive method, learning the image-text alignment from scratch while *using substantially less training resources and without any language co-training/pre-training*.
> This highlights the exceptional efficiency and promise of our end-to-end unified approach.
>
> **Performance Boost with Small-scale Supervised Fine-tuning:** To directly address the performance concern, we conducted a new experiment by adding a Supervised Fine-Tuning (SFT) stage.
> We curate a small dataset (around 100k images and 20k videos, **0.2\% of the pre-training data**) for SFT, which aligns with human aesthetic preference and follows closely with the text instructions.
> We fine-tune our Lumos-1 1B and 3B on this small-scale data.
> We list the performance below on GenEval and VBench after SFT.
>
> |Method|GenEval Overall|Single Obj.|Two Obj.|Counting|Colors|Position | Attr. Bind|
> |:-|:-:|:-:|:-:|:-:|:-:|:-:|:-:|
> |MMaDA 8B|0.63|0.99|0.76|0.61|0.84|0.20|0.37|
> |EMU3 8B |0.66|0.99|0.81|0.42|0.80|0.49|0.45|
> | Lumos-1 1B |0.601|0.959|0.732|0.375|0.774|0.365|0.400|
> | Lumos-1 3B |0.664|0.953|0.806|0.463|0.806|0.483|0.475|
> | Lumos-1 1B (SFT) |0.725|0.984|0.869|0.519|0.862|0.558|0.558|
> | Lumos-1 3B (SFT) |**0.777**|**0.991**|**0.897**|**0.566**|**0.878**|**0.640**|**0.690**|
>
> |Method|Semantic Score|Quality Score|VBench Overall|
> |:-|:-:|:-:|:-:|
> |OpenSoraPlan (2.7B + 13B)|65.62|**80.14**|77.23|
> |Lumos-1 1B|68.65|78.27|76.34|
> |Lumos-1 3B|73.51|79.52|78.32|
> |Lumos-1 1B (SFT)|72.45|79.60|78.17|
> |Lumos-1 3B (SFT)|**77.53**|79.80|**79.35**|
>
>
> The SFT stage dramatically boosts performance on GenEval, with our 3B model achieving a score of 0.777, significantly outperforming much larger models like EMU3 and MMaDA.
> The massive gains in challenging metrics like Position and Attribute Binding show that SFT effectively aligns our model's powerful generative capability with complex human instructions.
> On VBench, the SFT stage also clearly improves the semantic score, which further confirms that Lumos-1 is highly effective at understanding and executing on nuanced prompts after a small amount of alignment.
> Note that we find that SFT focuses on frame quality improvement, while the I2V task has the given first image; therefore, we do not see clear improvement on the I2V task, which is a future work to explore.
>
>
> **Performance Boost with a Diffusion Head:** To prove that our architecture's latent space is powerful and compatible with SOTA continuous decoders, we conducted a new training-free experiment.
> We take the output of Lumos-1 and use it to initialize a pre-trained diffusion model (Wan 1.3B) at a high noise step (0.98), then allow it to denoise for the final steps. This leverages Lumos-1's strong semantic guidance and Wan's high-fidelity continuous decoder.
>
> |Model|GenEval|
> |:-|:-:|
> |Lumos-1 3B|0.66|
> |Wan 1.3B|0.71|
> |Lumos-1 + Wan enhancing|0.73|
>
> |Model|Semantic Score|Quality Score|VBench Overall|
> |:-|:-:|:-:|:-:|
> |Lumos-1 3B|73.51|79.52|78.32|
> |Wan 1.3B|78.95|83.11|82.27|
> |Lumos-1 + Wan enhancing|80.14|83.07|82.48|
>
> This simple, training-free combination surpasses the standalone diffusion model on both GenEval and VBench, particularly in semantic scores. This shows: Lumos-1 produces a semantically superior latent structure that can be paired with better diffusion decoders.
> We believe that with large-scale training, the performance can be further boosted.
>
> **Other possible solutions** include:
> *1)* Integrating advanced tokenizers.
> *2)* Incorporating language co-training enhances the model's linguistic abilities.
>
>
> **Reference**:
> [1] Mogao: An Omni Foundation Model for Interleaved Multi-Modal Generation.

---

### Official Review · Reviewer_gCo6 · 2025-11-01

**Soundness:** 2
**Presentation:** 3
**Contribution:** 2
**Rating:** 4
**Confidence:** 5

**Summary:**

The paper presents Lumos-1, a unified autoregressive model for video generation using a discrete diffusion process. It combines text and video vocabularies and proposes two main techniques: MM-RoPE for better spatiotemporal position encoding, and AR-DF (Autoregressive Discrete Diffusion Forcing) to handle loss imbalance during masked diffusion training. The model is tested on T2I, I2V, and T2V tasks, showing comparable results to larger diffusion-based systems despite using fewer resources.

**Strengths:**

1. The study of RoPE frequency allocation is genuinely interesting. The proposed MM-RoPE makes sense — it fixes some imbalance in 3D RoPE and doesn’t really cost extra compute.
2. The setup is efficient: 60M images and 10M videos, trained on 48 H20 GPUs.
3. The experiments are complete, covering main benchmarks (T2I, T2V, I2V) and solid ablations.

**Weaknesses:**

1. The biggest weakness is the performance. Even though the paper claims efficiency, Lumos-1 doesn’t really beat diffusion or existing AR models on any benchmark. The visuals still look blurry and the motion often feels weird or distorted. It’s not convincing that this setup improves things beyond simplicity.
2. The experimental part only uses validation loss to validate different component like AR-DF and MM-RoPE. Including benchmark metrics would better demonstrate effectiveness since validation loss does not always aligns with down stream performance, even for AR models.
3. The frequency allocation for MRoPE is actually getting more and more attention nowadays. For example, [1] proposes to use U-RoPE. A concurrent work [2] gives a much more in-depth discussion about this, including HoPE and IL-RoPE. This paper only compares with VideoRoPE, weaken its contribution.
4. AR-DF looks like a combination of discrete diffusion and diffusion forcing, only with tube masking. But tube masking actually breaks the unidirectional dependency in AR models. This could hurt the scalability of the model.

[1] UniViT: Unifying Image and Video Understanding in One Vision Encoder
[2] Revisiting Multimodal Positional Encoding in Vision–Language Models

**Questions:**

Please refer to the weaknesses.

---

> ### Author Response · Authors · 2025-11-25
> **Response to Reviewer gCo6: Part I**
>
> Thank you for your insightful review.
> We're glad you found merit in the novelty of MM-RoPE, the efficiency of training Lumos-1, and the comprehensive experiments and ablation studies on three visual generation tasks.
> We have carefully considered your questions and hope our responses below provide the necessary clarification.
> Please do not hesitate to ask if anything is unclear. (W denotes weakness, Q denotes question, and A denotes answer.)
>
> > W1. The model shows simplicity but the performance does not stand out.
>
> **W1A1**: We appreciate that you mentioned the simplicity of our framework.
> One of the reasons that leads to the performance and the blurry issue is attributed to the discrete tokenizers, which usually trail continuous representations, rather than a failure of the unified architecture itself.
> We adopt discrete tokenizers in Lumos-1 to *align with the language tokens*, showing that using the standard cross-entropy loss can lead to a model with competitive performance.
> We believe that, with the advancement of research on discrete tokenizers, the performance of Lumos-1 can be largely boosted, which is beyond the scope of this work.
>
> To enable a fairer comparison, we add a column in Table 1, 2 and 3 to indicate the representation type used for generation, whether it is continuous or discrete.
> Moreover, we added more methods (public in 2025) using discrete tokenizers in Table 1.
> Compared with these methods like EMU3 8B and MMaDA 8B, Lumos-1 is still a competitive method, learning the image-text alignment from scratch while *using substantially less training resources and without any language co-training/pre-training*.
> This highlights the efficiency and promise of our end-to-end unified approach.
>
> **Performance Boost with Small-scale Supervised Fine-tuning:** To directly address the performance concern, we conducted a new experiment by adding a Supervised Fine-Tuning (SFT) stage.
> We curate a small dataset (around 100k images and 20k videos, **0.2\% of the pre-training data**) for SFT, which aligns with human aesthetic preference and follows closely with the text instructions.
> We fine-tune our Lumos-1 1B and 3B on this small-scale data.
> We list the performance below on GenEval and VBench after SFT.
>
> |Method|GenEval Overall|Single Obj.|Two Obj.|Counting|Colors|Position | Attr. Bind|
> |:-|:-:|:-:|:-:|:-:|:-:|:-:|:-:|
> |MMaDA 8B|0.63|0.99|0.76|0.61|0.84|0.20|0.37|
> |EMU3 8B |0.66|0.99|0.81|0.42|0.80|0.49|0.45|
> | Lumos-1 1B |0.601|0.959|0.732|0.375|0.774|0.365|0.400|
> | Lumos-1 3B |0.664|0.953|0.806|0.463|0.806|0.483|0.475|
> | Lumos-1 1B (SFT) |0.725|0.984|0.869|0.519|0.862|0.558|0.558|
> | Lumos-1 3B (SFT) |**0.777**|**0.991**|**0.897**|**0.566**|**0.878**|**0.640**|**0.690**|
>
> |Method|Semantic Score|Quality Score|VBench Overall|
> |:-|:-:|:-:|:-:|
> |OpenSoraPlan (2.7B + 13B)|65.62|**80.14**|77.23|
> |Lumos-1 1B|68.65|78.27|76.34|
> |Lumos-1 3B|73.51|79.52|78.32|
> |Lumos-1 1B (SFT)|72.45|79.60|78.17|
> |Lumos-1 3B (SFT)|**77.53**|79.80|**79.35**|
>
> The SFT stage dramatically boosts performance on GenEval, with our 3B model achieving a score of 0.777, significantly outperforming much larger models like EMU3 and MMaDA.
> The massive gains in challenging metrics like Position and Attribute Binding show that SFT effectively aligns our model's powerful generative capability with complex human instructions.
> On VBench, the SFT stage also clearly improves the semantic score, which further confirms that Lumos-1 is highly effective at understanding and executing on nuanced prompts after a small amount of alignment.
> Note that we find that SFT focuses on frame quality improvement, while the I2V task has the given first image; therefore, we do not see clear improvement on the I2V task, which is a future work to explore.
>
> **Performance Boost with a Diffusion Head:** To prove that our architecture's latent space is powerful and compatible with SOTA continuous decoders, we conducted a new training-free experiment.
> We take the output of Lumos-1 and use it to initialize a pre-trained diffusion model (Wan 1.3B) at a high noise step (0.98), then allow it to denoise for the final steps. This leverages Lumos-1's strong semantic guidance and Wan's high-fidelity continuous decoder.
>
> |Model|GenEval|
> |:-|:-:|
> |Lumos-1 3B|0.66|
> |Wan 1.3B|0.71|
> |Lumos-1 + Wan enhancing|0.73|
>
> |Model|Semantic Score|Quality Score|VBench Overall|
> |:-|:-:|:-:|:-:|
> |Lumos-1 3B|73.51|79.52|78.32|
> |Wan 1.3B|78.95|83.11|82.27|
> |Lumos-1 + Wan enhancing|80.14|83.07|82.48|
>
> This simple, training-free combination surpasses the standalone diffusion model on both GenEval and VBench, particularly in semantic scores. This shows Lumos-1 produces a semantically superior latent structure that can be paired with better diffusion decoders.
> We believe that with large-scale training, the performance can be further boosted.
>
> **Other possible solutions** include:
> *1)* Integrating advanced tokenizers.
> *2)* Incorporating language co-training enhances the model's linguistic abilities.

---

> ### Author Response · Authors · 2025-11-25
> **Response to Reviewer gCo6: Part II**
>
> > W2. Quantitative ablation studies on AR-DF and MM-RoPE.
>
> **W2A2**: Thank you for this valuable suggestion.
> We agree with the reviewer that comprehensive quantitative ablation studies are useful to demonstrate the efficacy of our proposed techniques.
>
> We would like to respectfully clarify that these studies were **included in the original submission** as Table 4 and Table 5 using GenEval and VBench metrics.
> We observe that *1)* eliminating both components would impact the performance; *2)* AR-DF is a better training scheme than other alternatives; *3)* both an appropriate scaling and the distributed frequency allocations contribute to the final boost.
>
> **Action Taken:**  We enlarged the table size of Table 4 and Table 5 to highlight their importance.
> We hope this clarification addresses your concern and are happy to provide further details if we misunderstood your question.
>
> > W3. Comparison with more RoPE variants.
>
> **W3A3**:
> Thank you for this suggestion. We agree that a detailed comparison with other RoPE variants is crucial. To address this, we have conducted new experiments comparing our MM-RoPE with the methods you mentioned, U-RoPE (online in Oct. 2025), IL-RoPE and HoPE.
> First of all, we compare the design philosophies under these RoPEs in the table below.
>
> | RoPE type | Compatiable with text RoPE | 3D structure | Comprehensive frequency allocation | Strategic scaling |
> |:-|:-:|:-:|:-:|:-:|
> | RoPE    |  ✔  |  ✗  |  ✗  |  ✗  |
> | M-RoPE [1]  |  ✔  |  ✔  |  ✗  |  ✗  |
> | U-RoPE [2]  |  ✔  |  ✔  |  ✗  |  ✗  |
> | IL-RoPE [3]  |  ✗  |  ✔  |  ✗  |  ✗  |
> | VideoRoPE [4] |  ✔  |  ✔  |  ✗  |  ✔  |
> | HoPE [5] |  ✔  |  ✔  |  ✗  |  ✔  |
> | MM-RoPE  |  ✔  |  ✔  |  ✔  |  ✔  |
>
> We mainly compare these methods on four aspects:
> *1)* their compatibility with text RoPE, so that models can align with contemporary LLMs;
> *2)* the inclusion of 3D information into RoPE to enable enhanced spatiotemporal modeling;
> *3)* the comprehensive frequency allocation that enables more balanced local-global context modeling;
> *4)* the strategic scaling of spatiotemporal indices (or their subsets) to account for different granularities of vision and text information.
> Based on the above analysis, MM-RoPE holistically incorporates the four properties.
>
> Moreover, we fine-tune Lumos-1 with the above-mentioned RoPE types and then quantitatively evaluate on GenEval and VBench (Overall Consistency and Imaging Quality) using a 1B model and 10k training following Table E3 of the Appendix.
>
> | RoPE Type | GenEval | Vbench-OC | Vbench-IQ |
> |:-|:-:|:-:|:-:|
> | M-RoPE    | 0.310 | 0.122 | 0.350 |
> | U-RoPE    | 0.402 | 0.165 | 0.423 |
> | IL-RoPE   | 0.541 | 0.225 | 0.513 |
> | VideoRoPE | 0.569 | 0.243 | 0.540 |
> | HoPE      | 0.570 | 0.246  |0.545 |
> | MM-RoPE   | 0.591 | 0.249 | 0.559 |
>
> We can observe from the table that:
> *1)* U-RoPE improves upon M-RoPE by reallocating high-frequency components for modeling local spatial details, which is useful;
> *2)* IL-RoPE allocating partial channels for modeling spatial information with an interleaved design, while it only allocates partial channels for modeling texts, hurting the performance;
> *3)* HoPE and VideoRoPE are similar due to their compatibility with text RoPE and their scaling for temporal indices, while HoPE improves upon VideoRoPE with a zero-frequency strategy for temporal modeling.
> Compared with them, MM-RoPE ensures text RoPE compatibility, improves frequency allocation that uses comprehensive frequency spectra for modeling spatiotemporal information and incorporates spatiotemporal scaling for vision-language alignment and visual resolution increasing.
>
> **Action Taken:** In the revised manuscript, we have expanded our discussions in the Related Work section and expanded experiments and analysis in Table 7 of the main paper (short version) and Appendix E2 (detailed version).

---

> > ### Author Response · Authors · 2025-12-03
> > **Response to Reviewer gCo6: Part III**
> >
> > > W4. Tube masking actually breaks the unidirectional dependency in AR models. This could hurt the scalability of the model.
> >
> > **W4A4**: Many thanks for this question. AR-DF indeed builds upon discrete diffusion and diffusion forcing, but furthermore, extends this paradigm to align more closely with the autoregressive nature of videos.
> > The core misunderstanding seems to be about the role of tube masking. We'd like to clarify that AR-DF is specifically *designed to preserve, not break, the unidirectional dependency*.
> >
> > In the original diffusion forcing, noise is applied independently to each frame, which implicitly ignores the correlation between consecutive frames.
> > In AR-DF, we take this dependency property into consideration by enabling the frame-level noise to be correlated.
> > Therefore, this design improvement reduces information leakage and ensures a better video modeling through balanced learning in every frame, which *does not break the unidirectional dependency*.
> > As shown in Table 4 of the original manuscript (also presented below), our strategy can outperform the original diffusion forcing, especially in video metrics.
> >
> > | Training Methods | GenEval | Vbench-OC | Vbench-IQ |
> > |:-|:-:|:-:|:-:|
> > | Diffusion Forcing | 0.590 | 0.241 | 0.540 |
> > | AR-DF | 0.591 | 0.249 | 0.559 |
> >
> > Regarding scalability, our experiments show that this is not a concern.
> > We observe consistent performance gains as we scale Lumos-1 to larger sizes on standard text-to-image, text-to-video, and image-to-video benchmarks.
> > This indicates that AR-DF does not impede, and in fact supports, the model's scalability.
> >
> > We are happy to provide further details if we misunderstood your question.
> >
> > **Reference**:
> > [1] Qwen2.5-VL Technical Report.
> > [2] UniViT: Unifying Image and Video Understanding in One Vision Encoder.
> > [3] Mogao: An Omni Foundation Model for Interleaved Multi-Modal Generation.
> > [4] VideoRoPE: What Makes for Good Video Rotary Position Embedding?
> > [5] HoPE: Hybrid of Position Embedding for Length Generalization in Vision-Language Models.

---

### Comment · Area_Chair_XmiP · 2025-11-27

Dear Reviewers,

This is a gentle reminder to please take a moment to review the author rebuttals and check whether your main concerns have been adequately addressed.

If possible, please update your reviews or add a brief clarification on whether the responses resolved your questions or if any issues remain. Your follow-up feedback is important for ensuring a fair and well-informed decision process.

Thank you again for your time and for helping maintain the quality of the ICLR review process.

Best,
AC

---

### Author Response · Authors · 2025-12-03
**Summary of the Rebuttal Phase: Part I**

Dear ACs, SACs and PCs,

As the rebuttal period comes to an end, we would like to present a brief summary of our rebuttal and provide the ACs with objective information for making the final decision.

**Summary of Our Work:** Our work introduces Lumos-1, a truly unified and efficient LLM-based framework for autoregressive video generation. We propose MM-RoPE for better spatiotemporal modeling using comprehensive frequency allocations and scaled 3D positions. We also introduce AR-DF for effective training and inference that solves the critical information leakage issue in mask-based discrete diffusion for videos. The obtained model Lumos-1 achieves very competitive results on T2I, I2V and T2V benchmarks while training from scratch and using limited resources.




------
***Reviewer's Consensus on Strengths***

First of all, we sincerely thank all reviewers for their efforts and valuable comments.
We are encouraged that the reviewers found the following merits of our work:
- `the novelty and usefulness of MM-RoPE` (Reviewer gCo6, NXEV, uiBr, J6ru),
- `the novelty of AR-DF` (Reviewer NXEV),
- `the rigor of designing MM-RoPE` (Reviewer J6ru),
- `the model training efficiency` (Reviewer gCo6),
- `comprehensive ablation studies and experiments on three generative tasks` (Reviewer gCo6, uiBr, J6ru) and
- `the significance of Lumos-1 in the era of unified models` (Reviewer NXEV, uiBr, J6ru).

We summarize our rebuttal below for your reference.

------
***Common Concerns from Reviewer gCo6 and NXEV***

**1. Regarding Comparisons with More RoPE Methods.**

We follow reviewers' suggestions to compare MM-RoPE with more RoPE methods, including U-RoPE, IL-RoPE and HoPE (VideoRoPE and M-RoPE were included in the original version).

- *Regarding the design philosophy*, MM-RoPE is the only method that are simultaneously compatible with text RoPE, has a 3D structure to encode spatiotemporal information, has comprehensive frequency allocation for spatiotemporal information and has a strategic spatiotemporal scaling for vision-language alignment and visual resolution increasing.

- *Regarding the quantitative performance*, we perform joint fine-tuning with the same setting and compare their corresponding performance using GenEval and VBench metrics, which further demonstrate the efficacy of MM-RoPE.

*Action Taken:* Experiments are added to Section 4.3 of the revised paper and Appendix E.2.


**2. Regarding the Performance Concern.**

We solve it by conducting a small-scale supervised fine-tuning (**using around 0.2\% of the pre-training data**), which clearly boosts the performance of Lumos 1B and 3B on **GenEval (0.601 -> 0.725, 0.664 -> 0.777, SOTA even compared with much larger models) and VBench (76.34 -> 78.17, 78.32 -> 79.35)**.
Moreover, we also demonstrate the use of an additional training-free diffusion head (Wan 1.3B) could boost the performance as well.
Both experiments show the potential of Lumos-1 in achieving better generation results.

*Action Taken:* Experiments are added to the revised paper.

------
***Additional Responses to Each Reviewer***

Apart from the above-mentioned concerns, we list the reviewer's major concerns and our responses below.

**1. Reviewer gCo6 (Original Score: 4)**

*Major concern:* 1. Quantitative ablation studies on AR-DF and MM-RoPE. 2. Tube masking actually breaks the unidirectional dependency.

*Our response:*
1. We apologize for the initial lack of clarity. This is a misunderstanding due to our small table sizes. The results are in Table 4 and 5 of the main paper, which we highlight after revision.

2. We do not break the unidirectional dependency due to the temporal causal masks we use and the specific consideration of temporal correlation in designing the temporal tube masks.


**2. Reviewer NXEV (Original Score: 4)**

*Major concern:* 1. Reconsidering the masking strategy of AR-DF as data augmentation. 2. Out-of-date results.

*Our response:*
1. The masking strategy is not data augmentation since data augmentation aims to add stochasticity to the training data, while AR-DF imposes a specific structure on the training to ensure the autoregressive nature of videos.
2. We update the results of Show-o and NOVA with up-to-date results.

**Note on Score Improvement:** Reviewer NXEV clearly indicates in the official review that they would raise the score if concerns are addressed.

---

> ### Author Response · Authors · 2025-12-03
> **Summary of the Rebuttal Phase: Part II**
>
> **3. Reviewer uiBr (Original Score: 6)**
>
> *Major concern:* 1. The possible training efficiency issue. 2. Comparison with other unified generative models or LLM-based autoregressive models. 3. The reason for not using LLM weights. 4. Analysis of the trade-off between semantic consistency and visual quality.
>
> *Our response:*
> 1. We compare with several methods in terms of the training computing, which shows the efficiency of training Lumos-1, while obtaining better performance. The results are placed in Appendix E.2.
> 2. We compare with DALL·E, LLamaGen, Loong and Lumina-mGPT in terms of architecture, position embedding, prediction paradigm, generated media and performance, showing that Lumos-1 has a truly unified architecture, has advanced positional encoding and has an efficient prediction paradigm. The results are updated in Appendix A.
> 3. We use Llama-2 architectures and tokenizers, which do not have 1B and 3B sizes.
> 4. The unified discrete codebook ensures semantic consistency and the performance of the discrete visual tokenizer determines the visual quality. Augmenting the visual tokenizer could achieve both benefits rather than a tradeoff.
>
>
>
> **4. Reviewer J6ru (Original Score: 6)**
>
> *Major concern:* 1. Provide a concrete discussion/roadmap to better align the title with the evidence. 2. Add content before fourth and fifth paragraphs. 3. Add more baselines in 2025. 4. The effect of AR-DF in long-form generation.
>
> *Our response:*
> 1. We provide a concrete roadmap to train a native unified model, with MM-RoPE and AR-DF used as key techniques. It is added to Appendix F.
> 2. We add and revise content in the third paragraph to better present the existing challenges to ensure smooth idea flow.
> 3. We add VideoMAR (NeurIPS 2025), SANA-1.5 (ICML 2025), Meissonic (ICLR 2025), Muddit (ICLR 2026 submission), UniDisc (arXiv 2025), D-DiT (CVPR 2025), MMaDA (NeurIPS 2025), Lumina-mGPT 2.0 (arXiv 2025) and Show-o2 (NeurIPS 2025) to the revised paper to show the effectiveness of Lumos-1.
> 4. To extend to long-form generation, we could use a sliding-window approach, without affecting shot transitions and dynamics.
>
> ------
>
> **In summary**, the reviewers have a consensus on the novelty of our core contributions (MM-RoPE, AR-DF), the comprehensiveness of our experiments and the significance of our work for the future of unified models.
> The primary concerns lie in performance and methodological comparisons.
> We have addressed all the concerns in the revised manuscript and the rebuttal.
> We firmly believe that Lumos-1 is a critical step towards better visual generation in the era of unified models.
>
> Your sincerely,
> Authors of Lumos-1

---

### Meta-Review · Area_Chair_Z98Q · 2026-01-07

**Summary:**

The reviewers generally agree that the paper presents solid and well-motivated technical contributions, particularly MM-RoPE and AR-DF, and that the experimental study is careful and substantially strengthened during rebuttal. Key concerns around novelty of positional encoding, missing comparisons, outdated baselines, training stability, and efficiency were largely addressed with additional experiments, clearer analysis, and updated results.

However, some reservations remain. The base model performance without supervised fine-tuning is still only competitive rather than clearly strong, and AR-DF, while empirically effective, lacks deeper theoretical justification. In addition, the “unified model” claim is mainly supported from a generation perspective, with understanding capabilities discussed via a roadmap rather than demonstrated empirically.

Overall, the rebuttal resolves most major technical concerns, but the remaining limitations keep the contribution incremental rather than decisive. On balance, I lean toward a weak accept, as the paper offers useful ideas and evidence that are likely to be valuable to the community, despite not being fully conclusive on all fronts.

**Reviewer Concerns:**

Most reviewer concerns were addressed in the rebuttal, but not all to the same extent.

MM-RoPE novelty and comparisons (NXEV, gCo6): Addressed. The authors added direct comparisons with IL-RoPE, U-RoPE, HoPE, and VideoRoPE, which resolves the novelty and missing-baseline concerns.

Lack of quantitative ablations (gCo6): Addressed. The rebuttal clarifies that GenEval and VBench metrics were already used and highlights these results more clearly.

Outdated or unfair baselines (NXEV): Addressed. Baselines were updated to recent results and reorganized for fairer comparison.

Training stability, efficiency, and architectural comparison (uiBr): Addressed with clearer explanations, additional compute analysis, and explicit structural comparisons.

Causality and scalability of AR-DF (gCo6): Largely addressed. The rebuttal clarifies the use of causal masks and provides empirical evidence that scalability is not harmed.

Outstanding or partially addressed concerns

Overall generation performance without SFT (gCo6): Still a concern. While SFT and diffusion-head results improve performance, the base model remains only competitive.

Theoretical justification of AR-DF (NXEV): Partially addressed. The method is empirically validated, but deeper theoretical grounding is still limited.

“Unified model” claim beyond generation (J6ru): Partially addressed. A roadmap is provided, but no direct understanding-task evidence is shown.

Overall, the rebuttal resolves most technical and comparison-related issues, with remaining concerns mainly about scope and depth rather than correctness.

**Reviewer Scores:**

Reviewer Scores

Reviewer gCo6
Original: 4
Expected after discussion: 4
Rationale: Core performance concerns remain only partially mitigated (base model still not clearly strong without SFT). No explicit signal from the reviewer that these concerns were fully resolved or that the score would increase.

Reviewer NXEV
Original: 4
Expected after discussion: 6
Rationale: The reviewer explicitly stated they would raise the score if concerns were addressed. The rebuttal directly added IL-RoPE / U-RoPE / HoPE comparisons, updated baselines, and clarified AR-DF, which reasonably removes the main blockers.

Reviewer uiBr
Original: 6
Expected after discussion: 6
Rationale: Major concerns (training stability, efficiency, architectural comparison, semantic vs. visual trade-off) were addressed, but the reviewer did not indicate a clear willingness to raise the score beyond the initial borderline accept.

Reviewer J6ru
Original: 6
Expected after discussion: 6
Rationale: Most issues were addressed, but the “unified model” concern was resolved via roadmap rather than evidence. No explicit signal that this would justify a higher score.

Summary
Expected scores: [4, 6, 6, 6]
Average expected score: 5.5

---

### Decision · Program_Chairs · 2026-01-26

Accept (Poster)